# SOLVING MIN-MAX OPTIMIZATION WITH HIDDEN STRUCTURE VIA GRADIENT DESCENT ASCENT

## ABSTRACT

Many recent AI architectures are inspired by zero-sum games, however, the behavior of their dynamics is still not well understood. Inspired by this, we study standard gradient descent ascent (GDA) dynamics in a specific class of non-convex non-concave zero-sum games, that we call hidden zero-sum games. In this class, players control the inputs of smooth but possibly non-linear functions whose outputs are being applied as inputs to a convex-concave game. Unlike general zero-sum games, these games have a well-defined notion of solution; outcomes that implement the von-Neumann equilibrium of the "hidden" convex-concave game. We prove that if the hidden game is strictly convex-concave then vanilla GDA converges not merely to local Nash, but typically to the von-Neumann solution. If the game lacks strict convexity properties, GDA may fail to converge to any equilibrium, however, by applying standard regularization techniques we can prove convergence to a von-Neumann solution of a slightly perturbed zero-sum game. Our convergence guarantees are non-local, which as far as we know is a first-of-its-kind type of result in non-convex non-concave games. Finally, we discuss connections of our framework with generative adversarial networks.

## 1 INTRODUCTION

Traditionally, our understanding of convex-concave games revolves around von Neumann's celebrated minimax theorem, which implies the existence of saddle point solutions with a uniquely defined value. Although many learning algorithms are known to be able to compute such saddle points (Cesa-Bianchi & Lugoisi, 2006), recently there has there has been a fervor of activity in proving stronger results such as faster regret minimization rates or analysis of the day-to-day behavior (Mertikopoulos et al., 2018; Daskalakis et al., 2018; Bailey & Piliouras, 2018; Abernethy et al., 2018; Wang & Abernethy, 2018; Daskalakis & Panageas, 2019; Abernethy et al., 2019; Mertikopoulos et al., 2019; Bailey & Piliouras, 2019; Gidel et al., 2019; Zhang & Yu, 2019; Hsieh et al., 2019; Bailey et al., 2020; Mokhtari et al., 2020; Hsieh et al., 2020; Pérolat et al., 2020).

This interest has been largely triggered by the impressive successes of AI architectures inspired by min-max games such as Generative Adversarial Networks (GANS) (Goodfellow et al., 2014a), adversarial training (Madry et al., 2018) and reinforcement learning self-play in games (Silver et al., 2017). Critically, however, all these applications are based upon *non-convex non-concave games*, our understanding of which is still nascent. Nevertheless, some important early work in the area has focused on identifying new solution concepts that are widely applicable in general min-max games, such as (local/differential) Nash equilibrium (Adolphs et al., 2019; Mazumdar & Ratliff, 2019), local minmax (Daskalakis & Panageas, 2018), local minimax (Jin et al., 2019), (local/differential) Stackleberg equilibrium (Fiez et al., 2020), local robust point (Zhang et al., 2020). The plethora of solutions concepts is perhaps suggestive that "solving" general min-max games unequivocally may be too ambitious a task. Attraction to spurious fixed points (Daskalakis & Panageas, 2018), cycles (Vlatakis-Gkaragkounis et al., 2019), robustly chaotic behavior (Cheung & Piliouras, 2019; Cheung & Piliouras, 2020) and computational hardness issues (Daskalakis et al., 2020) all suggest that general min-max games might inherently involve messy, unpredictable and complex behavior.

**Are there rich classes of non-convex non-concave games with an effectively unique game theoretic solution that is selected by standard optimization dynamics (e.g. gradient descent)?**

**Our class of games.** We will define a general class of min-max optimization problems, where each agent selects its own vectors of parameters which are then processed separately by smooth functions. Each agent receives their respective payoff after entering the outputs of the processed decision vectors as inputs to a standard convex-concave game. Formally, there exist functions $\mathbf{F} : \mathbb{R}^N \to X \subset \mathbb{R}^n$ and $\mathbf{G} : \mathbb{R}^M \to Y \subset \mathbb{R}^m$ and a continuous convex-concave function $L : X \times Y \to \mathbb{R}$, such that the min-max game is

$$\min_{\boldsymbol{\theta} \in \mathbb{R}^N} \max_{\boldsymbol{\phi} \in \mathbb{R}^M} L(\mathbf{F}(\boldsymbol{\theta}), \mathbf{G}(\boldsymbol{\phi})). \qquad \text{(Hidden Convex-Concave (HCC))}$$

We call this class of min-max problems Hidden Convex-Concave Games. It generalizes the recently defined hidden bilinear games of Vlatakis-Gkaragkounis et al. (2019).

**Our solution concept.** Out of all the local Nash equilibria of HCC games, there exists a special subclass, the vectors $(\boldsymbol{\theta}^*, \boldsymbol{\phi}^*)$ that implement the von Neumann solution of the convex-concave game. This solution has a strong and intuitive game theoretic justification. Indeed, it is stable even if the agents could perform arbitrary deviations directly on the output spaces $X, Y$. These parameter combinations $(\boldsymbol{\theta}^*, \boldsymbol{\phi}^*)$ "solve" the "hidden" convex-concave $L$ and thus we call them *von Neumann solutions*. Naturally, HCCs will typically have numerous local saddle/Nash equilibria/fixed points that do not satisfy this property. Instead, they correspond to stationary points of the $\mathbf{F}, \mathbf{G}$ where their output is stuck, e.g., due to an unfortunate initialization. At these points the agents may be receiving payoffs which can be arbitrarily smaller/larger than the game theoretic value of game $L$. Fortunately, we show that Gradient Descent Ascent (GDA) strongly favors von Neumann solutions over generic fixed points.

**Our results.** In this work, we study the behavior of continuous GDA dynamics for the class of HCC games where each coordinate of $\mathbf{F}, \mathbf{G}$ is controlled by disjoint sets of variables. In a nutshell, we show that GDA trajectories stabilize around or converge to the corresponding von Neumann solutions of the hidden game. Despite restricting our attention to a subset of HCC games, our analysis has to overcome unique hurdles not shared by standard convex concave games.

*Challenges of HCC games.* In convex-concave games, deriving the stability of the von Neumann solutions relies on the Euclidean distance from the equilibrium being a Lyapunov function. In contrast, in HCC games where optimization happens in the parameter space of $\boldsymbol{\theta}, \boldsymbol{\phi}$, the non-linear nature of $\mathbf{F}, \mathbf{G}$ distorts the convex-concave landscape in the output space. Thus, the Euclidean distance will not be in general a Lyapunov function. Moreover, the existence of *any* Lyapunov function for the trajectories in the output space of $\mathbf{F}, \mathbf{G}$ does not translate to a well-defined function in the parameter space (unless $\mathbf{F}, \mathbf{G}$ are trivial, invertible maps). Worse yet, even if $L$ has a unique solution in the output space, this solution could be implemented by multiple equilibria in the parameter space and thus each of them can not be individually globally attracting. Clearly any transfer of stability or convergence properties from the output to the parameter space needs to be initialization dependent.

*Lyapunov Stability.* Our first step is to construct an initialization-dependent Lyapunov function that accounts for the curvature induced by the operators $\mathbf{F}$ and $\mathbf{G}$ (Lemma 2). Leveraging a potentially infinite number of initialization-dependent Lyapunov functions in Theorem 4 we prove that under mild assumptions the outputs of $\mathbf{F}, \mathbf{G}$ stabilize around the von Neumann solution of $L$.

*Convergence.* Mirroring convex concave games, we require strict convexity or concavity of $L$ to provide convergence guarantees to von Neumann solutions (Theorem 5). Barring initializations where von Neumann solutions are not reachable due to the limitations imposed by $\mathbf{F}$ and $\mathbf{G}$, the set of von Neumann solutions are globally asymptotically stable (Corollary 1). Even in non-strict HCC games, we can add regularization terms to make $L$ strictly convex concave. Small amounts of regularization allows for convergence without significantly perturbing the von Neumann solution (Theorem 6) while increasing regularization enables exponentially faster convergence rates (Theorem 7).

**Organization.** In Section 2 we provide some preliminary notation, the definition of our model and some useful technical lemmas. Section 3 is devoted to the presentation of our the main results. Section 4 discusses applications of our framework to specific GAN formulations. Section 5 concludes our work with a discussion of future directions and challenges. We defer the full proofs of our results as well as further discussion on applications to the Appendix.

## 2 PRELIMINARIES

### 2.1 NOTATION

Vectors are denoted in boldface $\mathbf{x}, \mathbf{y}$ unless otherwise indicated are considered as column vectors. We use $\|\cdot\|$ corresponds to denote the $\ell_2-$norm. For a function $f : \mathbb{R}^d \to \mathbb{R}$ we use $\nabla f$ to denote its gradient. For functions of two vector arguments, $f(\mathbf{x}, \mathbf{y}) : \mathbb{R}^{d_1} \times \mathbb{R}^{d_2} \to \mathbb{R}$ , we use $\nabla_{\mathbf{x}} f, \nabla_{\mathbf{y}} f$ to denote its partial gradient. For the time derivative we will use the dot accent abbreviation, i.e., $\dot{\mathbf{x}} = \frac{d}{dt}[\mathbf{x}(t)]$. A function $f$ will belong to $C^r$ if it is $r$ times continuously differentiable. The term "sigmoid" function refers to $\sigma : \mathbb{R} \to \mathbb{R}$ such that $\sigma(x) = (1 + e^{-x})^{-1}$.

### 2.2 HIDDEN CONVEX CONCAVE GAMES

We will begin our discussion by defining the notion of convex concave functions as well as strictly convex concave functions. Note that our definition of strictly convex concave functions is a superset of strictly convex strictly concave functions that are usually studied in the literature.

**Definition 1.** *$L : \mathbb{R}^n \times \mathbb{R}^m \to \mathbb{R}$ is convex concave if for every $\mathbf{y} \in \mathbb{R}^n$ $L(\cdot, \mathbf{y})$ is convex and for every $\mathbf{x} \in \mathbb{R}^m$ $L(\mathbf{x}, \cdot)$ is concave. Function $L$ will be called strictly convex concave if it is convex concave and for every $\mathbf{x} \times \mathbf{y} \in \mathbb{R}^n \times \mathbb{R}^m$ either $L(\cdot, \mathbf{y})$ is strictly convex or $L(\mathbf{x}, \cdot)$ is strictly concave.*

At the center of our definition of HCC games is a convex concave utility function $L$. Additionally, each player of the game is equipped with a set of operator functions. The minimization player is equipped with $n$ functions $f_i : \mathbb{R}^{n_i} \to \mathbb{R}$ while the maximization player is equipped with $m$ functions $g_j : \mathbb{R}^{m_j} \to \mathbb{R}$. We will assume in the rest of our discussion that $f_i, g_j, L$ are all $C^2$ functions. The inputs $\boldsymbol{\theta}_i \in \mathbb{R}^{n_i}$ and $\boldsymbol{\phi}_j \in \mathbb{R}^{m_j}$ are grouped in two vectors

$$\boldsymbol{\theta} = [\boldsymbol{\theta}_1 \quad \boldsymbol{\theta}_2 \quad \cdots \quad \boldsymbol{\theta}_n]^\top \qquad \mathbf{F}(\boldsymbol{\theta}) = [f_1(\boldsymbol{\theta}_1) \quad f_2(\boldsymbol{\theta}_2) \quad \cdots \quad f_N(\boldsymbol{\theta}_n)]^\top$$
$$\boldsymbol{\phi} = [\boldsymbol{\phi}_1 \quad \boldsymbol{\phi}_2 \quad \cdots \quad \boldsymbol{\phi}_m]^\top \qquad \mathbf{G}(\boldsymbol{\theta}) = [g_1(\boldsymbol{\phi}_1) \quad g_2(\boldsymbol{\phi}_2) \quad \cdots g_M(\boldsymbol{\phi}_m)]^\top$$

We are ready to define the hidden convex concave game

$$(\boldsymbol{\theta}^*, \boldsymbol{\phi}^*) = \arg\min_{\boldsymbol{\theta} \in \mathbb{R}^N} \arg\max_{\boldsymbol{\phi} \in \mathbb{R}^M} L(\mathbf{F}(\boldsymbol{\theta}), \mathbf{G}(\boldsymbol{\phi})).$$

where $N = \sum_{i=1}^n n_i$ and $M = \sum_{j=1}^m m_j$. Given a convex concave function $L$, all stationary points of $L$ are (global) Nash equilibria of the min-max game. We will call the set of all equilibria of $L$, von Neumann solutions of $L$ and denote them by Solution($L$). Unfortunately, Solution($L$) can be empty for games defined over the entire $\mathbb{R}^n \times \mathbb{R}^m$. For games defined over convex compact sets, the existence of at least one solution is guaranteed by von Neumann's minimax theorem. Our definition of HCC games can capture games on restricted domains by choosing appropriately bounded functions $f_i$ and $g_j$. In the following sections, we will just assume that Solution($L$) is not empty. We note that our results hold for both bounded and unbounded $f_i$ and $g_j$. We are now ready to write down the equations of the GDA dynamics for a HCC game:

$$\dot{\boldsymbol{\theta}}_i = -\nabla_{\boldsymbol{\theta}_i} L(\mathbf{F}(\boldsymbol{\theta}), \mathbf{G}(\boldsymbol{\phi})) = -\nabla_{\boldsymbol{\theta}_i} f_i(\boldsymbol{\theta}_i) \frac{\partial L}{\partial f_i}(\mathbf{F}(\boldsymbol{\theta}), \mathbf{G}(\boldsymbol{\phi}))$$

$$\dot{\boldsymbol{\phi}}_j = \nabla_{\boldsymbol{\phi}_j} L(\mathbf{F}(\boldsymbol{\theta}), \mathbf{G}(\boldsymbol{\phi})) = \nabla_{\boldsymbol{\phi}_j} g_j(\boldsymbol{\phi}_j) \frac{\partial L}{\partial g_j}(\mathbf{F}(\boldsymbol{\theta}), \mathbf{G}(\boldsymbol{\phi}))$$

(1)

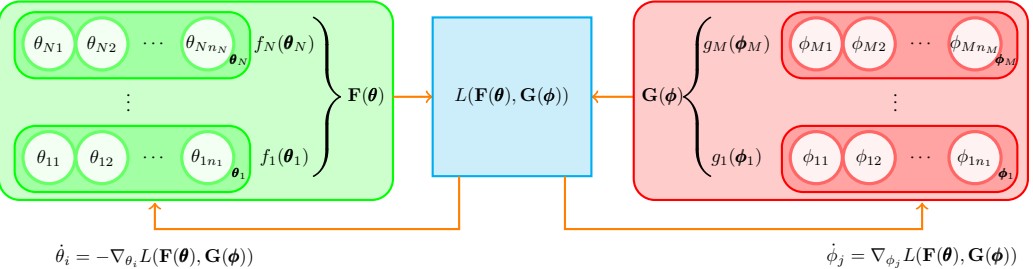

## 2.3 REPARAMETRIZATION

The following lemma is useful in studying the dynamics of hidden games.

**Lemma 1.** *Let $k : \mathbb{R}^d \to \mathbb{R}$ be a $C^2$ function. Let $h : \mathbb{R} \to \mathbb{R}$ be a $C^1$ function and $\mathbf{x}(t)$ denote the unique solution of the dynamical system $\Sigma_1$. Then the unique solution for dynamical system $\Sigma_2$ is $\mathbf{z}(t) = \mathbf{x}(\int_0^t h(s)\mathrm{d}s)$*

$$\left.\begin{cases} \dot{\mathbf{x}} &=& \nabla k(\mathbf{x}) \\ \mathbf{x}(0) &=& \mathbf{x}_{init} \end{cases}\right\} : \Sigma_1 \qquad \left.\begin{cases} \dot{\mathbf{z}} &=& h(t)\nabla k(\mathbf{z}) \\ \mathbf{z}(0) &=& \mathbf{x}_{init} \end{cases}\right\} : \Sigma_2 \qquad (2)$$

Figure 1: Neither Gradient Descent nor Ascent can traverse stationary points. An immediate consequence of Lemma 1 is that if we initialize in the above example $\theta_i(0)$ at (a), $f_i(\theta_i(t))$ can not escape the purple section. This extends to cases where $\boldsymbol{\theta}_i$ is vector of variables.

By choosing $h(t) = -\partial L(\mathbf{F}(t), \mathbf{G}(t))/\partial f_i$ and $h(t) = \partial L(\mathbf{F}(t), \mathbf{G}(t))/\partial g_j$ respectively, we can connect the dynamics of each $\boldsymbol{\theta}_i$ and $\boldsymbol{\phi}_j$ under Equation (1) to gradient ascent on $f_i$ and $g_j$. Applying Lemma 1, we get that trajectories of $\boldsymbol{\theta}_i$ and $\boldsymbol{\phi}_j$ under Equation (1) are restricted to be subsets of the corresponding gradient ascent trajectories with the same initializations. For example, in Figure 1 $\theta_i(t)$ can not escape the purple section if it is initialized at (a) neither the orange section if it is initialiazed at (f). This limits the attainable values that $f_i(t)$ and $g_j(t)$ can take for a specific initialization. Let us thus define the following:

**Definition 2.** *For each initialization $\mathbf{x}(0)$ of $\Sigma_1$, $\mathrm{Im}_k(\mathbf{x}(0))$ is the image of $k \circ \mathbf{x} : \mathbb{R} \to \mathbb{R}$.*

Applying Definition 2 in the above example, $\mathrm{Im}_{f_i}(\theta_i(0)) = (f_i(-2), f_i(-1))$ if $\theta_i$ is initialized at (c). Additionally, observe that in each colored section $f_i(\theta_i(t))$ uniquely identifies $\theta_i(t)$. Generally, even in the case that $\boldsymbol{\theta}_i$ are vectors, Lemma 1 implies that for a given $\boldsymbol{\theta}_i(0)$, $f_i(\boldsymbol{\theta}_i(t))$ uniquely identifies $\boldsymbol{\theta}_i(t)$. As a result we get that a new dynamical system involving only $f_i$ and $g_j$

**Theorem 1.** *For each initialization $(\boldsymbol{\theta}(0), \boldsymbol{\phi}(0))$ of Equation (1), there are $C^1$ functions $X_{\boldsymbol{\theta}_i(0)}$, $X_{\boldsymbol{\phi}_j(0)}$ such that $\boldsymbol{\theta}_i(t) = X_{\boldsymbol{\theta}_i(0)}(f_i(t))$ and $\boldsymbol{\phi}_j(t) = X_{\boldsymbol{\phi}_j(0)}(g_j(t))$. If $(\boldsymbol{\theta}(t), \boldsymbol{\phi}(t))$ satisfy Equation (1) then $f_i(t) = f_i(\boldsymbol{\theta}_i(t))$ and $g_j(t) = g_j(\boldsymbol{\phi}_j(t))$ satisfy*

$$\dot{f}_i = -\|\nabla_{\boldsymbol{\theta}_i} f_i(X_{\boldsymbol{\theta}_i(0)}(f_i))\|^2 \frac{\partial L}{\partial f_i}(\mathbf{F}, \mathbf{G})$$

$$\dot{g}_j = \|\nabla_{\boldsymbol{\phi}_j} g_j(X_{\boldsymbol{\phi}_j(0)}(g_j))\|^2 \frac{\partial L}{\partial g_j}(\mathbf{F}, \mathbf{G})$$

$$(3)$$

By determining the ranges of $f_i$ and $g_j$, an initialization clearly dictates if a von Neumann solution is attainable. In Figure 1 for example, any point of the pink, orange or blue colored section like (e), (f) or (g) can not converge to a von Neumann solution with $f_i(\theta_i) = f_i^*$. The notion of *safety* captures which initializations can converge to a given element of Solution($L$).

**Definition 3.** . *We will call the initialization $(\boldsymbol{\theta}(0), \boldsymbol{\phi}(0))$ safe for a $(\mathbf{p}, \mathbf{q}) \in$ Solution($L$) if $\boldsymbol{\phi}_i(0)$ and $\boldsymbol{\theta}_j(0)$ are not stationary points of $f_i$ and $g_j$ respectively and $p_i \in \mathrm{Im}_{f_i}(\boldsymbol{\theta}_i(0))$ and $q_j \in \mathrm{Im}_{g_j}(\boldsymbol{\phi}_j(0))$.*

Finally, in the following sections we use some fundamental notions of stability. We call an equilibrium $\mathbf{x}^*$ of an autonomous dynamical system $\dot{\mathbf{x}} = \mathcal{D}(\mathbf{x}(t))$ *stable* if for every neighborhood $U$ of $\mathbf{x}^*$ there is a neighborhood $V$ of $\mathbf{x}^*$ such that if $\mathbf{x}(0) \in V$ then $\mathbf{x}(t) \in U$ for all $t \geq 0$. We call a set $S$ *asymptotically stable* if there exists a neighborhood $\mathcal{R}$ such that for any initialization $\mathbf{x}(0) \in \mathcal{R}$, $\mathbf{x}(t)$ approaches $S$ as $t \to +\infty$. If $\mathcal{R}$ is the whole space the set *globally asymptotically stable*.

## 3 LEARNING IN HIDDEN CONVEX CONCAVE GAMES

### 3.1 GENERAL CASE

Our main results are based on designing a Lyapunov function for the dynamics of Equation (3):

**Lemma 2.** *If $L$ is convex concave and $(\boldsymbol{\phi}(0), \boldsymbol{\theta}(0))$ is a safe for $(\mathbf{p}, \mathbf{q}) \in Solution(L)$, then the following quantity is non-increasing under the dynamics of Equation* (3)*:*

$$H(\mathbf{F}, \mathbf{G}) = \sum_{i=1}^{N} \int_{p_i}^{f_i} \frac{z - p_i}{\|\nabla f_i(X_{\boldsymbol{\theta}_i(0)}(z))\|^2} \mathrm{d}z + \sum_{j=1}^{M} \int_{q_j}^{g_j} \frac{z - q_j}{\|\nabla g_j(X_{\boldsymbol{\phi}_j(0)}(z))\|^2} \mathrm{d}z \qquad (4)$$

Observe that our Lyapunov function here is not the distance to $(\mathbf{p}, \mathbf{q})$ as in a classical convex concave game. The gradient terms account for the non constant multiplicative terms in Equation (3). Indeed if the game was not hidden and $f_i$ and $g_j$ were the identity functions then $H$ would coincide with the Euclidean distance to $(\mathbf{p}, \mathbf{q})$. Our first theorem employs the above Lyapunov function to show that $(\mathbf{p}, \mathbf{q})$ is stable for Equation (3).

**Theorem 2.** *If $L$ is convex concave and $(\boldsymbol{\phi}(0), \boldsymbol{\theta}(0))$ is a safe for $(\mathbf{p}, \mathbf{q}) \in Solution(L)$, then $(\mathbf{p}, \mathbf{q})$ is stable for Equation* (3)*.*

Clearly, for the special case of globally invertible functions $\mathbf{F}, \mathbf{G}$ we could come up with an equivalent Lyapunov function in the $\boldsymbol{\theta}, \boldsymbol{\phi}$-space. In this case it is straightforward to transfer the stability results from the induced dynamical system of $\mathbf{F}, \mathbf{G}$ (Equation (3)) to the initial dynamical system of $\boldsymbol{\theta}, \boldsymbol{\phi}$ (Equation (1)). For example we can prove the following result:

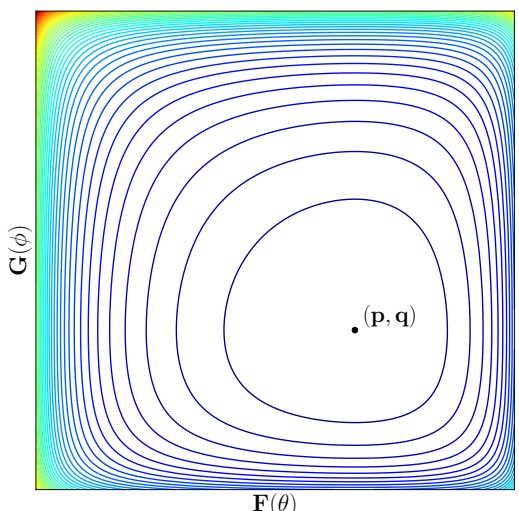

Figure 2: Level sets of Lyapunov function of Equation (4) for both $\mathbf{F}$ and $\mathbf{G}$ being one dimensional sigmoid functions.

**Theorem 3.** *If $f_i$ and $g_j$ are sigmoid functions and $L$ is convex concave and there is a $(\boldsymbol{\phi}(0), \boldsymbol{\theta}(0))$ that is safe for $(\mathbf{p}, \mathbf{q}) \in Solution(L)$, then $(\mathbf{F}^{-1}(\mathbf{p}), \mathbf{G}^{-1}(\mathbf{q}))$ is stable for Equation* (1)*.*

In the general case though, stability may not be guaranteed in the parameter space of Equation (1). We will instead prove a weaker notion of stability, which we call hidden stability. Hidden stability captures that if $(\mathbf{F}(\boldsymbol{\theta}(0)), \mathbf{G}(\boldsymbol{\phi}(0)))$ is close to a von Neumann solution, then $(\mathbf{F}(\boldsymbol{\theta}(t)), \mathbf{G}(\boldsymbol{\phi}(t)))$ will remain close to that solution. Even though hidden stability is weaker, it is essentially what we are interested in, as the output space determines the utility that each player gets. Here we provide sufficient conditions for hidden stability.

**Theorem 4** (Hidden Stability). *Let $(\mathbf{p}, \mathbf{q}) \in Solution(L)$. Let $R_{f_i}$ and $R_{g_j}$ be the set of regular values[1] of $f_i$ and $g_j$ respectively. Assume that there is a $\xi > 0$ such that $[p_i - \xi, p_i + \xi] \subseteq R_{f_i}$ and $[q_j - \xi, q_j + \xi] \subseteq R_{g_j}$. Define*

$$r(t) = \|\mathbf{F}(\boldsymbol{\theta}(t)) - \mathbf{p}\|^2 + \|\mathbf{G}(\boldsymbol{\phi}(t)) - \mathbf{q}\|^2.$$

*If $f_i$ and $g_j$ are proper functions[2], then for every $\epsilon > 0$, there is an $\delta > 0$ such that*

$$r(0) < \delta \implies \forall t \geq 0 : r(t) < \epsilon.$$

Unfortunately hidden stability still does not imply convergence to von Neumann solutions. Vlatakis-Gkaragkounis et al. (2019) studied hidden bilinear games and proved that $\dot{H} = 0$ for this special class of HCC games. Hence, a trajectory is restricted to be a subset of a level set of $H$ which is bounded away from the equilibrium as shown in Figure 2. To sidestep this, we will require in the next subsection the hidden game to be strictly convex concave.

---

[1] A value $a \in \operatorname{Im} f$ is called a regular value of $f$ if $\forall q \in \operatorname{dom} f : f(q) = a$, it holds $\nabla f(q) \neq \mathbf{0}$.
[2] A function is proper if inverse images of compact subsets are compact.

## 3.2 Hidden strictly convex concave games

In this subsection we focus on the case where $L$ is a strictly convex concave function. Based on Definition 1, a strictly convex concave game is not necessarily strictly convex strictly concave and thus it may have a continuum of von Neumann solutions. Despite this, LaSalle's invariance principle, combined with the strict convexity concavity, allows us to prove that if $(\boldsymbol{\theta}(0), \boldsymbol{\phi}(0))$ is safe for $Z \subseteq \text{Solution}(L)$ then $Z$ is locally asymptotically stable for Equation (3).

**Lemma 3.** *Let $L$ be strictly convex concave and $Z \subset \text{Solution}(L)$ is the non empty set of equilbria of $L$ for which $(\boldsymbol{\theta}(0), \boldsymbol{\phi}(0))$ is safe. Then $Z$ is locally asymptotically stable for Equation (3).*

The above lemma however does not suffice to prove that for an arbitrary initialization $(\boldsymbol{\theta}(0), \boldsymbol{\phi}(0))$, $(\mathbf{F}(t), \mathbf{G}(t))$ approaches $Z$ as $t \to +\infty$. In other words, *a-priori* it is unclear if $(\mathbf{F}(\boldsymbol{\theta}(0)), \mathbf{G}(\boldsymbol{\phi}(0)))$ is necessarily inside the region of attraction (ROC) of $Z$. To get a refined estimate of the ROC of $Z$, we analyze the behavior of $H$ as $f_i$ and $g_j$ approach the boundaries of $\text{Im}_{f_i}(\boldsymbol{\theta}_i(0))$ and $\text{Im}_{g_j}(\boldsymbol{\phi}_j(0))$ and more precisely we show that the level sets of $H$ are bounded. Once again the corresponding analysis is trivial for convex concave games, since the level sets are spheres around the equilibria.

**Theorem 5.** *Let $L$ be strictly convex concave and $Z \subset \text{Solution}(L)$ is the non empty set of equilbria of $L$ for which $(\boldsymbol{\theta}(0), \boldsymbol{\phi}(0))$ is safe. Under the dynamics of Equation (1) $(\mathbf{F}(\boldsymbol{\theta}(t)), \mathbf{G}(\boldsymbol{\theta}(t)))$ converges to a point in $Z$ as $t \to \infty$.*

The theorem above guarantees convergence to a von Neumann solution for all initializations that are safe for at least one element of $\text{Solution}(L)$. However, this is not the same as global asymptotic stability. To get even stronger guarantees, we can assume that all initializations are safe. In this case it is straightforward to get a global asymptotic stability result:

**Corollary 1.** *Let $L$ be strictly convex concave and assume that all intitializations are safe for at least one element of Solution(L). The following set is globally asymptotically stable for continuous GDA dynamics.*

$$\{(\boldsymbol{\theta}^*, \boldsymbol{\phi}^*) \in \mathbb{R}^n \times \mathbb{R}^m : (F(\boldsymbol{\theta}^*), G(\boldsymbol{\phi}^*)) \in Solution(L)\}$$

Notice that the above approach on global asymptotic convergence using Lyapunov arguments can be extended to other popular alternative gradient-based heuristics like variations of Hamiltonian Gradient descent. For concision, we defer the exact statements, proofs in Section 8.2.2

## 3.3 Convergence via regularization

Regularization is a key technique that works both in the practice of GANs Mescheder et al. (2018); Kurach et al. (2019) and in the theory of convex concave games Pérolat et al. (2020); Roth et al. (2017); Sanjabi et al. (2018). Our settings of hidden convex concave games allows for provable guarantees for regularization in a wide class of settings, bringing closer practical and theoretical guarantees. Let us have a utility $L(\mathbf{x}, \mathbf{y})$ that is convex concave but not strictly. Here we will propose a modified utility $L'$ that is strictly convex strictly concave. Specifically we will choose

$$L'(\mathbf{x}, \mathbf{y}) = L(\mathbf{x}, \mathbf{y}) + \frac{\lambda}{2}\|\mathbf{x}\|^2 - \frac{\lambda}{2}\|\mathbf{y}\|^2$$

The choice of the parameter $\lambda$ captures the trade-off between convergence to the original equilibrium of $L$ and convergence speed. On the one hand, invoking the implicit function theorem, we get that for small $\lambda$ the equilibria of $L$ are not significantly perturbed.

**Theorem 6.** *If $L$ is a convex concave function with invertible Hessians at all its equilibria, then for each $\epsilon > 0$ there is a $\lambda > 0$ such that $L'$ has equilibria that are $\epsilon$-close to the ones of $L$.*

Note that invertibility of the Hessian means that $L$ must have a unique equilibrium. On the other hand increasing $\lambda$ increases the rate of convergence of safe initializations to the perturbed equilibrium

**Theorem 7.** *Let $(\boldsymbol{\theta}(0), \boldsymbol{\phi}(0))$ be a safe initialization for the unique equilibrium of $L'$ $(\mathbf{p}, \mathbf{q})$. If*

$$r(t) = \|\mathbf{F}(\boldsymbol{\theta}(t)) - \mathbf{p}\|^2 + \|\mathbf{G}(\boldsymbol{\phi}(t)) - \mathbf{q}\|^2$$

*then there are initialization dependent constants $c_0, c_1 > 0$ such that $r(t) \leq c_0 \exp(-\lambda c_1 t)$.*

# 4 APPLICATIONS

In this section we show how our theorems provide connections between the framework of hidden games and practical applications of min-max optimization like training GANs.

**Hidden strictly convex-concave games**. We will start our discussion with the fundamental generative architecture of Goodfellow et al. (2014a)'s GAN. In the *vanilla GAN* architecture, as it is commonly referred, our goal is to find a generator distribution $p_G$ that is close to an input data distribution $p_{data}$. To find such a generator function, we can use a discriminator $D$ that "criticizes" the deviations of the generator from the input data distribution. For the case of a discrete $p_{data}$ over a set $\mathcal{N}$, the minimax problem of Goodfellow et al. (2014a) is the following:

$$\min_{\substack{p_G(x) \geq 0, \\ \sum_{x \in \mathcal{N}} p_G(x) = 1}} \max_{D \in (0,1)^{|\mathcal{N}|}} V(G, D) = \sum_{x \in \mathcal{N}} p_{data}(x) \log(D(x)) + \sum_{x \in \mathcal{N}} p_G(x) \log(1 - D(x))$$

The problem above can be formulated as a constrained strictly convex-concave hidden game. On the one hand, for a fixed discriminator $D^*$, the $V(G, D^*)$ is linear over the $p_G(x)$. On the other hand, for a fixed generator $G^*$, $V(G^*, D)$ is strongly-concave. We can implement the inequality constraints on both the generator probabilities and discriminator using sigmoid activations. For the equality constraint $\sum_{x \in \mathcal{N}} p_G(x) = 1$ we can introduce a Langrange multiplier. Having effectively removed the constraints, we can see in Figure 3, the dynamics of Equation (1) converge to the unique equilibrium of the game, an outcome consistent with our results in Corollary 1. It is worth noting that while the Euclidean distance to the equilibrium is not monotonically decreasing, $H(t)$ is.

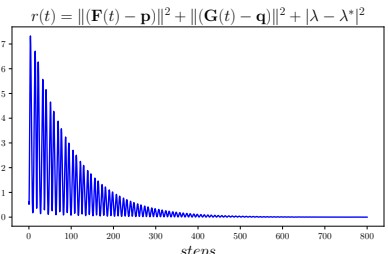
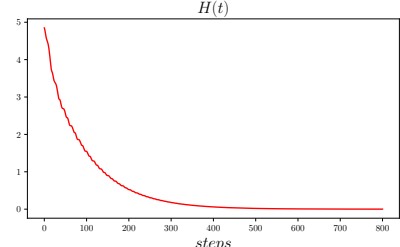

Figure 3: Comparison of $\ell_2$ distance from the equilibrium and the Lyapunov function. Both of them converge to zero as we state but $\ell_2$ distance not monotonically to zero. For $p_{data}$ we choose a fully mixed distribution of dimension $d = 4$. Given the sigmoid activations all the initializations are safe. We defer the detailed proof of convergence in Section 9.2.

**Hidden convex-concave games & Regularizaiton**. An even more interesting case is Wassertein GANs–WGANs (Arjovsky et al. (2017)). One of the contributions of Lei et al. (2019) is to show that WGANs trained with Stochastic GDA can learn the parameters of Gaussian distributions whose samples are transformed by non-linear activation functions. It is worth mentioning that the original WGAN formulation has a Lipschitz constraint in the discriminator function. For simplicity, Lei et al. (2019) replaced this constraint with a quadratic regularizer. The min-max problem for the case of one-dimensional Gaussian $\mathcal{N}(0, \alpha_*^2)$ and linear discriminator $D_v(x) = v^\top x$ with $x^2$ activation is:

$$\min_{\alpha \in \mathbb{R}} \max_{v \in \mathbb{R}} V_{WGAN}(G_\alpha, D_v) = \mathbb{E}_{\mathbf{X} \sim p_{data}}[D(\mathbf{X})] - \mathbb{E}_{\mathbf{X} \sim p_G}[D(\mathbf{X})] - v^2/2$$

$$= \mathbb{E}_{x \sim \mathcal{N}(0, \alpha_*^2)^2}[vx] - \mathbb{E}_{x \sim \mathcal{N}(0, \alpha_*^2)^2}[vx] - v^2/2$$

$$= (\alpha_*^2 - \alpha^2)v - v^2/2$$

Observe that $V_{WGAN}$ is not convex-concave but it can posed as a hidden strictly convex-concave game with $\mathbf{G}(\alpha) = (\alpha_*^2 - \alpha^2)$ and $\mathbf{F}(v) = v$. When computing expectations analytically without sampling, Theorem 5 guarantees convergence. In contrast, without the regularizer $V_{WGAN}$ can be modeled as a hidden linear-linear game and thus GDA dynamics cycle. Empirically, these results are robust to discrete and stochastic updates using sampling as shown in Figure 4. Therefore regularization in the work of Lei et al. (2019) was a vital ingredient in their proof strategy and not just an implementation detail. In Section 9.3, we also discuss applications of regularization to normal form zero sum games.

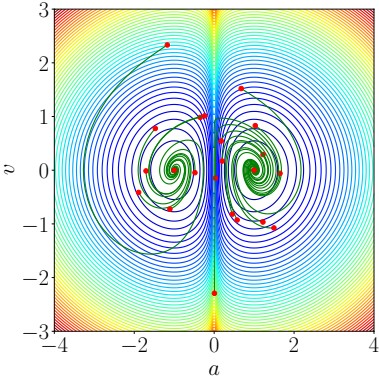 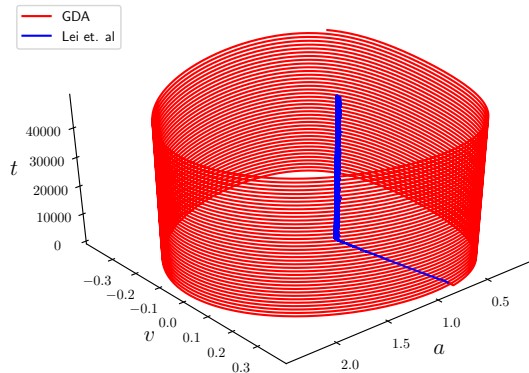

Figure 4: On the left, we show the trajectories of regularized GDA for $\alpha_*^2 = 1$ as well as the level sets of Equation (4). All trajectories converge to one of the two equilibria $(0, 1)$ and $(0, -1)$ whereas without regularization, GDA would cycle on the level sets. In the right figure, we replace the exact expectations in $V_{\text{WGAN}}$ with approximations via sampling and continuous time updates on $\alpha$ and $v$ with discrete ones. For small learning rates and large sample sizes, unregularized GDA continues to cycle. In contrast, the regularization approach of Lei et al. (2019) converges to the $(0, 1)$ equilibrium.

The two applications of HCC games in GANs are not isolated findings but instances of a broader pattern that connects HCC games and standard GAN formulations. As noted by Goodfellow (2017), if updates in GAN applications were directly performed in the "functional space", i.e. the generator and discriminator outputs, then standard arguments from convex concave optimization would imply convergence to global Nash equilibria. Indeed, standard GAN formulations like the vanilla GAN Goodfellow et al. (2014a), f-GAN (Nowozin et al. (2016)) and WGAN Arjovsky et al. (2017) can all be thought of as convex concave games in the space of generator and discriminator outputs. Given that the connections between convex concave games and standard GAN objectives in the output space is missing from recent literature, in Section 9.1 we show how one can apply Von Neumann's minimax theorem to derive the optimal generators and discriminators even in the non-realizable case. In practice, the updates happen in the parameter space and thus convexity arguments no longer apply. Our study of HCC games is a stepping stone towards bridging the gap in convergence guarantees between the case of direct updates in the output space and the parameter space.

## 5 DISCUSSION

In this work, we introduce a class of non-convex non-concave games that we call hidden convex concave (HCC) games. In this class of games, the competition on the output/operator space has a convex concave structure but training happens in the input/parameter space, where the mappings between input and output space are smooth but non-convex non-concave functions. The main inspiration for this class is the indirect competition of the parameters of generator and discriminator on GANs' architectures. Our analysis combines ideas from game theory, dynamical systems and control theory such Lyapunov functions and LaSalle's theorem. Our convergence results favor not arbitrary local Nash equilibria, but only von Neumann solutions. To the best of our knowledge, such last iterate convergence results are the first result of their kind. Given the modular structure of our model and proofs, HCC games show particular promise as a theoretical testbed for studying which dynamics are more well suited to which GANs. We believe that further positive results of this kind for different combinations of GAN formulations and learning algorithms are possible by properly adapting our current techniques.

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

CONTENTS

## 6 Background

### 6.1 Background in dynamical systems

Our analysis combines tools from dynamical systems, stability analysis and invariance principles theory. We start with the definitions of the different stability notions. We remind the well known Lyapunov's Lyapunov stability criterion (**Theorem 8**) Stability analysis in convex concave games is further complicated due to the possibility of non-isolated fixed points. To tackle this issue, we recall Krasovskii-LaSalle's Invariance Principle (**Theorem 9**), a powerful result that has several implications for the asymptotic stability of a set in an autonomous (possibly nonlinear) dynamical system. In the special case where the goal set contains only stable fixed points a pointwise convergence theorem can be derived (**Theorem 10**). Finally, we remind the notions of diffeomorphism and topological conjugacy of two dynamical systems, which are useful to transfer behavioral claims between equivalent dynamics.

Let $\mathbf{f} : D \to \mathbb{R}^n$ be a locally Lipschitz map from a domain $D \subset \mathbb{R}^n$ to $\mathbb{R}^n$. We consider dynamical systems of the form

$$\dot{\mathbf{x}} = \mathbf{f}(\mathbf{x}) \tag{$\star$}$$

A point $\bar{\mathbf{x}}$ for which $\mathbf{f}(\bar{\mathbf{x}}) = \mathbf{0}$ is called a fixed point. We will be interested in the following notions of stability for the fixed point points of Equation ($\star$).

**Definition 4** (Stability properties, (Khalil, 2002, Definition 4.1)). *The fixed point $\mathbf{x} = \mathbf{0}$ of Equation ($\star$) is*

- *stable if, for each $\epsilon > 0$, there is a $\delta = \delta(\epsilon) > 0$ such that*

$$\|\mathbf{x}(0)\| < \delta \implies \|\mathbf{x}(t)\| < \epsilon \quad \forall t \geq 0$$

- *unstable if it is not stable*

- *asymptotically stable if it is stable and $\delta$ can be chosen such that*

$$\|\mathbf{x}(0)\| < \delta \implies \lim_{t \to \infty} \mathbf{x}(t) = \mathbf{0}$$

The Lyapunov Theorem will be a useful tool to prove (asymptotic) stability of a fixed point.

**Theorem 8** (Lyapunov Theorem, (Khalil, 2002, Theorem 4.1)). *Let $\mathbf{x} = \mathbf{0}$ be a fixed point point for Equation ($\star$) and $D \subset \mathbb{R}^n$ be a domain containing $\mathbf{x} = \mathbf{0}$. Let $V : D \to \mathbb{R}$ be a continuously differentiable function such that*

$$V(\mathbf{0}) = 0 \text{ and } V(\mathbf{x}) > 0 \text{ in } D - \{\mathbf{0}\}$$
$$\dot{V}(\mathbf{x}) \leq 0 \text{ in } D$$

*then $\mathbf{x} = \mathbf{0}$ is stable. Moreover if*

$$\dot{V}(\mathbf{x}) < 0 \text{ in } D - \{\mathbf{0}\}$$

*then $\mathbf{x} = \mathbf{0}$ is asymptotically stable.*

Unfortunately, the Lyapunov theorem is not very helpful when it comes to proving convergence in dynamical systems with non isolated fixed points. By definition, non-isolated fixed points cannot be asymptotically stable. Non isolated fixed points may give rise to more complex behaviour than point-wise convergence.

**Definition 5.** *We say that a trajectory $\mathbf{x}(t)$ approaches a set $\mathcal{M}$ as $t \to \infty$ if for each $\epsilon > 0$ there is a $T > 0$ such that*

$$dist(\mathbf{x}(t), \mathcal{M}) < \epsilon, \quad \forall t > T$$

*where the operator "dist" is the minimum distance from a point to a set $\mathcal{M}$*

$$dist(\mathbf{p}, \mathcal{M}) = \inf_{\mathbf{x} \in \mathcal{M}} \|\mathbf{p} - \mathbf{x}\|$$

**Definition 6.** *We say that a set $\mathcal{M}$ is invariant for Equation ($\star$) if*

$$\mathbf{x}(0) \in \mathcal{M} \implies \mathbf{x}(t) \in \mathcal{M}, \quad \forall t \in \mathbb{R}$$

*We will say $\mathcal{M}$ is positively invariant if the above holds for $t \geq 0$.*

We are ready to state LaSalle's Invariance Principle, a general theorem that can help us study the stability of non isolated fixed points.

**Theorem 9** ( LaSalle's Invariance Principle, (Khalil, 2002, Theorem 4.4)). *Let $\Omega \subset D$ be a compact set that is positively invariant with respect to Equation ($\star$). Let $V : D \to \mathbb{R}$ be a continuously differentiable function such that $\dot{V}(\mathbf{x}) \leq 0$ in $\Omega$. Let $E$ be the set of all points where $\dot{V}(\mathbf{x}) = 0$. Let $\mathcal{M}$ be the largest invariant set in $E$. Then every solution starting in $\Omega$ approaches $\mathcal{M}$ as $t \to \infty$.*

LaSalle's theorem does not give us pointwise convergence directly. But in the special case that $\mathcal{M}$ contains only stable fixed points we can apply the following theorem

**Theorem 10** (Pointwise Convergence Theorem, (Bhat & Bernstein, 2003, Proposition 5.4)). *Let $\mathbf{x}(t)$ be a trajectory of Equation ($\star$). If the positive limit sets of $\mathbf{x}(t)$ contain a stable fixed point then $\mathbf{x}(t)$ converges to it as $t \to \infty$.*

**Definition 7** (Differomorphism, Perko (1991)). *Let $U, V$ be manifolds. A map $f : U \to V$ is called a diffeomorphism if $f$ carries $U$ onto $V$ and also both $f$ and $f^{-1}$ are smooth.*

**Definition 8** (Topological conjugacy, Perko (1991)). *Two flows $\Phi_t : A \to A$ and $\Psi_t : B \to B$ are conjugate if there exists a homeomorphism $g : A \to B$ such that*

$$\forall \boldsymbol{x} \in A, t \in \mathbb{R} : g(\Phi_t(\boldsymbol{x})) = \Psi_t(g(\boldsymbol{x}))$$

*Furthermore, two flows $\Phi_t : A \to A$ and $\Psi_t : B \to B$ are diffeomorphic if there exists a diffeomorphism $g : A \to B$ such that*

$$\forall \boldsymbol{x} \in A, t \in \mathbb{R} : g(\Phi_t(\boldsymbol{x})) = \Psi_t(g(\boldsymbol{x})).$$

*If two flows are diffeomorphic, then their vector fields are related by the derivative of the conjugacy. That is, we get precisely the same result that we would have obtained if we simply transformed the coordinates in their differential equations.*

## 6.2 BACKGROUND IN CONVEX OPTIMIZATION

> For the sake of completeness, we recall here the definition of (strict) convex/concave function and its first order necessary and sufficient criterion. We will also discuss strong convexity and its second order characterizations.

We will be interested in notions from convex optimization throughout this work

**Definition 9** ((Boyd & Vandenberghe, 2004, p. 67)). *Let $f : \mathbb{R}^n \to \mathbb{R}$ be a function then*

- *$f$ is convex if*

$$\forall \mathbf{x}, \mathbf{y} \in \mathbb{R}^n, t \in [0, 1] : f(t\mathbf{x} + (1 - t)\mathbf{y}) \leq tf(\mathbf{x}) + (1 - t)f(\mathbf{y})$$

- *$f$ is strictly convex if*

$$\forall \mathbf{x}, \mathbf{y} \in \mathbb{R}^n, t \in (0, 1) : f(t\mathbf{x} + (1 - t)\mathbf{y}) < tf(\mathbf{x}) + (1 - t)f(\mathbf{y})$$

- *$f$ is (strictly) concave if $-f$ is (strictly) convex.*

We will also use the first order characterizations of convex and concave functions

**Theorem 11** ((Boyd & Vandenberghe, 2004, p. 69-70)). *Let $f : \mathbb{R}^n \to \mathbb{R}$ be a differentiable function.*

- *$f$ is convex if and only if $\forall \mathbf{x}, \mathbf{y} \in \mathbb{R}^n : f(\mathbf{y}) \geq f(\mathbf{x}) + \nabla f(\mathbf{x})^T(\mathbf{y} - \mathbf{x})$*
- *$f$ is concave if and only if $\forall \mathbf{x}, \mathbf{y} \in \mathbb{R}^n : f(\mathbf{y}) \leq f(\mathbf{x}) + \nabla f(\mathbf{x})^T(\mathbf{y} - \mathbf{x})$*

To establish convergence rates, we will use the notion of strong convexity

**Definition 10** ((Nesterov, 2004, p. 63)). *A continuously differentiable function $f$ of $\mathbb{R}^n$ will be called $\mu$ strongly convex for a positive constant $\mu$ if for all $\mathbf{x}, \mathbf{y} \in \mathbb{R}^n$ we have*

$$f(\mathbf{y}) \geq f(\mathbf{x}) + \langle \nabla f(\mathbf{x}), \mathbf{y} - \mathbf{x} \rangle + \frac{\mu}{2} \|\mathbf{x} - \mathbf{y}\|^2$$

We will also use second order characterizations of strong convexity

**Theorem 12** ((Nesterov, 2004, p. 65)). *A twice continuously differentiable function $f$ is $\mu$ strongly convex for a positive constant $\mu$ if and only if for all $\mathbf{x} \in \mathbb{R}^n$ we have*

$$\nabla^2 f(\mathbf{x}) \geq \mu I$$

Symmetrically, a function will be called $\mu$ strongly concave if $-f$ is $\mu$ strongly convex.

## 6.3 BACKGROUND IN GAME THEORY

> In this short section, we remind to the reader a generalization of Von-Neumann's Minimax theorem, which we will exploit to analyze the equilibrium solution of the different GANs' architectures. A special case of Fan's minimax theorem is the following

**Corollary 2** (Fan's minimax theorem, Fan (1953)). *Let $X \subset \mathbb{R}^n$ and $Y \subset \mathbb{R}^m$ be convex non-empty sets. Suppose that $X$ is compact and $f : X \times Y \to \mathbb{R}$ is a function such that $f(\cdot, y)$ is lower semicontinuous on $X$ for each $y \in Y$ and that $f$ is convex concave. Then we have that*

$$\min_{x \in X} \sup_{y \in Y} f(x, y) = \sup_{y \in Y} \min_{x \in X} f(x, y).$$

## 7 PRELIMINARIES

> The below time-reparametrization lemma shows that the solution for a non-autonomous system, multiplicative to a gradient flow can be derived by just time-rescaling of the solution of the simplified gradient ascent dynamics. Indeed, since the multiplicative term is common across all terms of the vector field then over the time it dictates only the magnitude of the vector field (the speed of the motion), but does not affect the directionality other than moving backwards or forwards along the same trajectory.

**Lemma 1.** *Let $k : \mathbb{R}^d \to \mathbb{R}$ be a $C^2$ function. Let $h : \mathbb{R} \to \mathbb{R}$ be a $C^1$ function and $\mathbf{x}(t)$ denote the unique solution of the dynamical system $\Sigma_1$. Then the unique solution for dynamical system $\Sigma_2$ is $\mathbf{z}(t) = \mathbf{x}(\int_0^t h(s)\mathrm{d}s)$*

$$\left\{ \begin{array}{ccc} \dot{\mathbf{x}} & = & \nabla k(\mathbf{x}) \\ \mathbf{x}(0) & = & \mathbf{x}_{init} \end{array} \right\} : \Sigma_1 \quad \left\{ \begin{array}{ccc} \dot{\mathbf{z}} & = & h(t)\nabla k(\mathbf{z}) \\ \mathbf{z}(0) & = & \mathbf{x}_{init} \end{array} \right\} : \Sigma_2 \tag{5}$$

*Proof.* Firstly, notice that it holds $\mathbf{x}(0) = \mathbf{x}_{\text{init}}$ and $\dot{\mathbf{x}} = \nabla k(\mathbf{x})$, since $\mathbf{x}$ is the unique solution of $\Sigma_1$ It is easy to check that:

$$\mathbf{z}(0) = \mathbf{x}(\int_0^0 h(s)\mathrm{d}s) = \mathbf{x}(0) = \mathbf{x}_{\text{init}}$$

$$\dot{\mathbf{z}} = \dot{\mathbf{x}}\left(\int_0^t h(s)\mathrm{d}s\right) \times \frac{\mathrm{d}[\int_0^t h(s)\mathrm{d}s]}{\mathrm{d}t}$$

$$= \nabla k\left(\mathbf{x}\left(\int_0^t h(s)\mathrm{d}s\right)\right) h(t) = \nabla k(\mathbf{z})h(t)$$

$\square$

> In order to leverage the convex-concave properties of the operators in our hidden structure under the Gradient Descent Ascent dynamics we need to recover the equivalent system $(T)$ in the operator space $\left(\begin{smallmatrix}\dot{\mathbf{F}} \\ \dot{\mathbf{G}}\end{smallmatrix}\right) = T\left[\left(\begin{smallmatrix}\mathbf{F} \\ \mathbf{G}\end{smallmatrix}\right)\right]$.
>
> $$(\Sigma) := \left\{ \begin{array}{ccc} \dot{\boldsymbol{\theta}} & = & - & \nabla L(\mathbf{F}(\boldsymbol{\theta}), \mathbf{G}(\boldsymbol{\phi})) \\ \dot{\boldsymbol{\phi}} & = & & \nabla L(\mathbf{F}(\boldsymbol{\theta}), \mathbf{G}(\boldsymbol{\phi})) \end{array} \right\} \equiv \left\{ \begin{array}{ccc} \dot{\boldsymbol{\theta}}_i & = & - & \nabla_{\boldsymbol{\theta}_i} f_i(\boldsymbol{\theta}_i) h_{f_i,L}(t) \\ \dot{\boldsymbol{\phi}}_j & = & & \nabla_{\boldsymbol{\phi}_j} g_j(\boldsymbol{\phi}_j) h_{g_j,L}(t) \end{array} \right\}$$
>
> From this point, applying the aforementioned lemma, under GDA each $f_i$ and $g_j$ follows a time dependent rescaling of the corresponding gradient ascent solution. Exploiting the monotonicity of $f_i(t)$ and $g_j(t)$ under gradient ascent, we can construct an invertible map between the parameter space $\{(\boldsymbol{\theta}_i, \boldsymbol{\phi}_j)\}$ and the operator space $\{(f_i, g_j)\}$ which allows us to construct the equivalent system $T$ in the operator space. *Notice that the properties of gradient ascent are crucial since the operator space can be arbitrarily smaller in dimension. In this case a smooth invertible map that is common for all initializations cannot exist.*

**Theorem 1.** *For each initialization $(\boldsymbol{\theta}(0), \boldsymbol{\phi}(0))$ of Equation (1), there are $C^1$ functions $X_{\boldsymbol{\theta}_i(0)}$, $X_{\boldsymbol{\phi}_j(0)}$ such that $\boldsymbol{\theta}_i(t) = X_{\boldsymbol{\theta}_i(0)}(f_i(t))$ and $\boldsymbol{\phi}_j(t) = X_{\boldsymbol{\phi}_j(0)}(g_j(t))$. If $(\boldsymbol{\theta}(t), \boldsymbol{\phi}(t))$ satisfy Equation (1) then $f_i(t) = f_i(\boldsymbol{\theta}_i(t))$ and $g_j(t) = g_j(\boldsymbol{\phi}_j(t))$ satisfy*

$$\dot{f}_i = -\|\nabla_{\boldsymbol{\theta}_i} f_i(X_{\boldsymbol{\theta}_i(0)}(f_i))\|^2 \frac{\partial L}{\partial f_i}(\mathbf{F}, \mathbf{G})$$

$$\dot{g}_j = \|\nabla_{\boldsymbol{\phi}_j} g_j(X_{\boldsymbol{\phi}_j(0)}(g_j))\|^2 \frac{\partial L}{\partial g_j}(\mathbf{F}, \mathbf{G}) \tag{3}$$

*Proof.* Let us first study a simpler dynamical system $(\Sigma^*)$ with unique solution of $\gamma_{\boldsymbol{\theta}_i(0)}(t)$.

$$(\Sigma^*) \equiv \left\{ \begin{array}{ccc} \dot{\mathbf{z}} & = & \nabla f_i(\mathbf{z}) \\ \mathbf{z}(0) & = & \boldsymbol{\theta}_i(0) \end{array} \right\}$$

It is easy to observe that:

$$\dot{f}_i = \nabla f(\mathbf{z})\dot{\mathbf{z}} = \|\nabla f(\mathbf{z})\|^2$$

If $\boldsymbol{\theta}_i(0)$ is a stationary point of $f_i$ then the trajectory of $\mathbf{z}$ is a single point. But the trajectory of $\boldsymbol{\theta}_i$ under the dynamics of Equation (1) is also a single point so we can pick the following function

$$X_{\boldsymbol{\theta}_i(0)}(f_i) = \boldsymbol{\theta}_i(0).$$

On the other hand if $\boldsymbol{\theta}_i(0)$ is not a stationary point of $f_i$, $f_i$ continuously increases along the trajectory of $(\Sigma^*)$. Therefore $A_{\boldsymbol{\theta}_i(0)}(t) = f_i(\gamma_{\boldsymbol{\theta}_i(0)}(t))$ is an increasing function and therefore invertible. Let us call $A_{\boldsymbol{\theta}_i(0)}^{-1}(f_i)$ the inverse.

Let's recall now the $\boldsymbol{\theta}_i$ part of the dynamical system of interest Equation (1)

$$\dot{\boldsymbol{\theta}}_i = -\nabla_{\boldsymbol{\theta}_i} f_i(\boldsymbol{\theta}_i) \frac{\partial L}{\partial f_i}(\mathbf{F}(\boldsymbol{\theta}), \mathbf{G}(\boldsymbol{\phi}))$$

initialized at $\boldsymbol{\theta}_i(0)$. Applying Lemma 1 for the first equation with

$$h(t) = -\frac{\partial L}{\partial f_i}(\mathbf{F}(\boldsymbol{\theta}(t)), \mathbf{G}(\boldsymbol{\phi}(t)))$$

we have that under the dynamics of Equation (1)

$$\boldsymbol{\theta}_i(t) = \gamma_{\boldsymbol{\theta}_i(0)}\left(\int_0^t h(s)\mathrm{d}s\right) \tag{P}$$

Thus it holds

$$f_i(\boldsymbol{\theta}_i(t)) = f\left(\gamma_{\boldsymbol{\theta}_i(0)}\left(\int_0^t h(s)\mathrm{d}s\right)\right) = A_{\boldsymbol{\theta}_i(0)}\left(\int_0^t h(s)\mathrm{d}s\right)$$

or equivalently

$$\int_0^t h(s)\mathrm{d}s = A_{\boldsymbol{\theta}_i(0)}^{-1}(f_i(\boldsymbol{\theta}_i(t)))$$

Plugging in back to Equation (P)

$$\boldsymbol{\theta}_i(t) = \gamma_{\boldsymbol{\theta}_i(0)}(A_{\boldsymbol{\theta}_i(0)}^{-1}(f_i(\boldsymbol{\theta}_i(t))))$$

Therefore we can pick

$$X_{\boldsymbol{\theta}_i(0)}(f_i) = \gamma_{\boldsymbol{\theta}_i(0)}(A_{\boldsymbol{\theta}_i(0)}^{-1}(f_i))$$

which is $C^1$ as composition of $C^1$ functions. We can perform an equivalent analysis for $\boldsymbol{\phi}_j(0)$ and $g_j$ to pick $C^1$ function $X_{\boldsymbol{\phi}_j(0)}$. Let us now track the time derivative of $f_i(\boldsymbol{\theta}_i)$ and $g_j(\boldsymbol{\phi}_j)$

$$\dot{f}_i = \nabla_{\boldsymbol{\theta}_i} f_i(\boldsymbol{\theta}_i)\dot{\boldsymbol{\theta}}_i = \|\nabla_{\boldsymbol{\theta}_i} f_i(\boldsymbol{\theta}_i)\|^2 \frac{\partial L}{\partial f_i}(\mathbf{F}, \mathbf{G})$$

$$\dot{g}_j = \nabla_{\boldsymbol{\phi}_j} g_j(\boldsymbol{\phi}_j)\dot{\boldsymbol{\phi}}_j = \|\nabla_{\boldsymbol{\phi}_j} g_j(\boldsymbol{\phi}_j)\|^2 \frac{\partial L}{\partial g_j}(\mathbf{F}, \mathbf{G})$$

We can now replace $\boldsymbol{\theta}_i = X_{\boldsymbol{\theta}_i(0)}(f_i)$ and $\boldsymbol{\phi}_j = X_{\boldsymbol{\phi}_j(0)}(g_j)$ to get the equations required. $\qquad\square$

## 8    HIDDEN CONVEX CONCAVE GAMES

> In this section, we analyze the derived stability properties of the hidden convex concave games. It is worth mentioning that without strict/strong convexity/concavity from at least one of the operators, the quality of the results are limited to "Lyapunov Stability". Firstly, we present a construction of a Lyapunov function for the operators' dynamics **Theorem 2**. Then, in **Theorem 3, Theorem 4** we explore the stability of the initial conditions in the parameter space.

## 8.1 GENERAL CASE

The following theorem presents the construction of a Lyapunov potential function for the induced operator dynamics. To motivate its construction, we can study a fundamental convex-concave function $L(x, y) = (x-p)^2 - (y-q)^2$ with saddle point $(p, q)$. Under the gradient-descent-ascent dynamics

$$(T) := \begin{cases} \dot{x} & = & - & \nabla_x L(x, y) & \text{(minimization of convex part)} \\ \dot{y} & = & & \nabla_y L(x, y) & \text{(maximization of concave part)} \end{cases}.$$

it is easy to check that $H(x, y) = (x - p)^2 + (y - q)^2$ meets all the criteria of a Lyapunov function. The construction below extends this argument to any convex-concave function $L(\mathbf{F}, \mathbf{G})$ and bypasses the more complex multiplicative terms for the gradient induced dynamics of Theorem 1. *Notice that*

$$H(\mathbf{F}, \mathbf{G}) = \sum_{i=1}^{N} \int_{p_i}^{f_i} \frac{z - p_i}{\|\nabla f_i(X_{\boldsymbol{\theta}_i(0)}(z))\|^2} \mathrm{d}z + \sum_{j=1}^{M} \int_{q_j}^{g_j} \frac{z - q_j}{\|\nabla g_j(X_{\boldsymbol{\phi}_j(0)}(z))\|^2} \mathrm{d}z$$

*coincides with the $\ell_2^2$ distance from $(\mathbf{p}, \mathbf{q})$ in the case of gradient norms equal to one, i.e.*

$$\|\nabla f_i\|^2 = \|\nabla g_j\|^2 = 1$$

**Lemma 2.** *If $L$ is convex concave and $(\boldsymbol{\phi}(0), \boldsymbol{\theta}(0))$ is a safe for $(\mathbf{p}, \mathbf{q}) \in Solution(L)$, then the following quantity is non-increasing under the dynamics of Equation (3):*

$$H(\mathbf{F}, \mathbf{G}) = \sum_{i=1}^{N} \int_{p_i}^{f_i} \frac{z - p_i}{\|\nabla f_i(X_{\boldsymbol{\theta}_i(0)}(z))\|^2} \mathrm{d}z + \sum_{j=1}^{M} \int_{q_j}^{g_j} \frac{z - q_j}{\|\nabla g_j(X_{\boldsymbol{\phi}_j(0)}(z))\|^2} \mathrm{d}z \qquad (4)$$

*Proof.* Simple substitution gets us the following

$$\dot{H} = -\sum_{i=1}^{N} (f_i - p_i) \frac{\partial L}{\partial f_i}(\mathbf{F}, \mathbf{G}) + \sum_{j=1}^{M} (g_j - q_j) \frac{\partial L}{\partial g_j}(\mathbf{F}, \mathbf{G})$$

$$= -\langle \mathbf{F} - \mathbf{p}, \nabla_{\mathbf{F}} L(\mathbf{F}, \mathbf{G}) \rangle + \langle \mathbf{G} - \mathbf{q}, \nabla_{\mathbf{G}} L(\mathbf{F}, \mathbf{G}) \rangle$$

By Theorem 11 for the convex $L(\cdot, \mathbf{G})$ and concave $L(\mathbf{F}, \cdot)$.

$$-\langle \mathbf{F} - \mathbf{p}, \nabla_{\mathbf{F}} L(\mathbf{F}, \mathbf{G}) \rangle \le L(\mathbf{p}, \mathbf{G}) - L(\mathbf{F}, \mathbf{G})$$
$$\langle \mathbf{G} - \mathbf{q}, \nabla_{\mathbf{G}} L(\mathbf{F}, \mathbf{G}) \le L(\mathbf{F}, \mathbf{G}) - L(\mathbf{F}, \mathbf{q})$$

Thus we can end up writing

$$\dot{H} \le L(\mathbf{p}, \mathbf{G}) - L(\mathbf{F}, \mathbf{G}) + L(\mathbf{F}, \mathbf{G}) - L(\mathbf{F}, \mathbf{q})$$
$$\le L(\mathbf{p}, \mathbf{G}) - L(\mathbf{p}, \mathbf{q}) + L(\mathbf{p}, \mathbf{q}) - L(\mathbf{F}, \mathbf{q}) \le 0$$

The last inequality holds since $(\mathbf{p}, \mathbf{q}) \in Solution(L)$. Indeed, if $(\mathbf{p}, \mathbf{q})$ is a saddle point of $L$ then $L(\mathbf{p}, \mathbf{G}) \le L(\mathbf{p}, \mathbf{q}) \le L(\mathbf{F}, \mathbf{q})$. □

**Theorem 2.** *If $L$ is convex concave and $(\boldsymbol{\phi}(0), \boldsymbol{\theta}(0))$ is a safe for $(\mathbf{p}, \mathbf{q}) \in Solution(L)$, then $(\mathbf{p}, \mathbf{q})$ is stable for Equation (3).*

*Proof.* Leveraging Lemma 2, there is a function $H$ which is well defined in $D = \{\text{Im}_{f_i}(\boldsymbol{\theta}_i(0))\}_{i=1}^{N} \times \{\text{Im}_{g_j}(\boldsymbol{\phi}_j(0))\}_{j=1}^{M}$ and in this domain $\dot{H} \le 0$. Given the safety conditions we know that $(\mathbf{p}, \mathbf{q}) \in D$. Observe that for the proposed function, it holds that $H(\mathbf{p}, \mathbf{q}) = 0$. Also for each $f_i$ and $g_j$ term in $H$ we know that it has its minimum of value 0 at the corresponding $p_i$ and $q_j$. We can deduce this by taking the derivative of each term to study its monotonicity. For example, the $f_i$ terms are strictly increasing in $f_i > p_i$ and strictly decreasing in $f_i < p_i$. Thus for all $D - \{(\mathbf{p}, \mathbf{q})\}$, $H > 0$. Applying Theorem 8 for the continuously differentiable $H$ we have that $(\mathbf{p}, \mathbf{q})$ is stable for Equation (3). □

> In the following example, we examine how it is possible to transfer the stability properties between two (topological conjugate) dynamical systems.

**Theorem 3.** *If $f_i$ and $g_j$ are sigmoid functions and $L$ is convex concave and there is a $(\phi(0), \theta(0))$ that is safe for $(\mathbf{p}, \mathbf{q}) \in Solution(L)$, then $(\mathbf{F}^{-1}(\mathbf{p}), \mathbf{G}^{-1}(\mathbf{q}))$ is stable for Equation (1).*

*Proof.* Firstly, we recall the property of sigmoid's gradient:

$$\frac{\mathrm{d}\sigma(x)}{\mathrm{d}x} = \sigma(x)(1 - \sigma(x)).$$

Thus the transformed dynamical system in the operator space can be written as:

$$(T) := \begin{cases} \dot{f}_i &= & -\quad f_i^2(1-f_i)^2 \frac{\partial L}{\partial f_i}(\mathbf{F}, \mathbf{G}) \\ \dot{g}_j &= & \quad g_j^2(1-g_j)^2 \frac{\partial L}{\partial g_j}(\mathbf{F}, \mathbf{G}) \end{cases}$$

*Notice that*

1. *The dynamical system $(T)$ in the operator space is* *independent* *of the initial conditions. In fact, the dynamical system of $(T)$ and the one of Equation (1), called $(\Sigma)$ for short, are diffeomorphic for all initializations, not just a specific trajectory.*

2. *Since $(\theta(0), \phi(0))$ is safe, using Theorem 2 we get that $(\mathbf{p}, \mathbf{q})$ is stable for $(T)$.*

We would like to prove that for every open neighborhood $V$ of $(\mathbf{F}^{-1}(\mathbf{p}), \mathbf{G}^{-1}(\mathbf{q}))$ there exists an open neighborhood $U$ of $(\mathbf{F}^{-1}(\mathbf{p}), \mathbf{G}^{-1}(\mathbf{q}))$ such that

$$(\theta_{\text{init}}, \phi_{\text{init}}) \in U \implies \forall t \geq 0 : (\theta(t), \phi(t)) \in V.$$

Using the diffeomorphism $\gamma = \gamma_{\Sigma \to T}$ between GDA dynamics of $(\Sigma)$ and $(T)$, $\gamma(V)$ is an open neighborhood of $(\mathbf{p}, \mathbf{q})$ since $V$ is open and $\gamma((\mathbf{F}^{-1}(\mathbf{p}), \mathbf{G}^{-1}(\mathbf{q}))) \equiv (\mathbf{p}, \mathbf{q}) \in \gamma(V)$. By Item 2, since $(\mathbf{p}, \mathbf{q})$ is stable for $(T)$ there is an open neighborhood $\tilde{U}$ of $(\mathbf{p}, \mathbf{q})$ such that:

$$(\mathbf{F}_{\text{init}}, \mathbf{G}_{\text{init}}) \in \tilde{U} \implies \forall t \geq 0 : (\mathbf{F}(t), \mathbf{G}(t)) \in \gamma(V)$$

or equivalently

$$\gamma(\theta_{\text{init}}, \phi_{\text{init}}) \in \tilde{U} \implies \forall t \geq 0 : \gamma(\theta(t), \phi(t)) \in \gamma(V)$$

Indeed, using the inverse diffeomorphism $\gamma^{-1}$, we can establish that for $U = \gamma^{-1}(\tilde{U})$ it holds that

$$(\theta_{\text{init}}, \phi_{\text{init}}) \in U \implies \forall t \geq 0 : (\theta(t), \phi(t)) \in V$$

$\square$

> Until now, we have established the stability of a pair $(\mathbf{p}, \mathbf{q})$ for the induced dynamics $(T)$. By the construction of the induced dynamics, $(T)$ is coupled only with a very specific initial condition $(\theta_{\text{init}}, \phi_{\text{init}})$. In order to tackle the challenge of a stability result for a whole region of initial conditions, in the following lemma we prove that $r(\theta, \phi) = \|\mathbf{F}(\theta) - \mathbf{p}\|^2 + \|\mathbf{G}(\phi) - \mathbf{q}\|^2$ can work like an intrinsic measure of closeness for the $\{\theta, \phi\}$-parameter space around a hidden fixed point of the $\{\mathbf{F}, \mathbf{G}\}$-operator space. Under this "hidden" neighborhood notion, stability property can be taken by assuming the properness of the hidden operators.

**Theorem 4.** *Let $(\mathbf{p}, \mathbf{q}) \in Solution(L)$. Let $R_{f_i}$ and $R_{g_j}$ be the set of regular values[3] of $f_i$ and $g_j$ respectively. Assume that there is a $\xi > 0$ such that $[p_i - \xi, p_i + \xi] \subseteq R_{f_i}$ and $[q_j - \xi, q_j + \xi] \subseteq R_{g_j}$. Define*

$$r(t) = \|\mathbf{F}(\theta(t)) - \mathbf{p}\|^2 + \|\mathbf{G}(\phi(t)) - \mathbf{q}\|^2.$$

*If $f_i$ and $g_j$ are proper functions[4], then for every $\epsilon > 0$, there is an $\delta > 0$ such that*

$$r(0) < \delta \implies \forall t \geq 0 : r(t) < \epsilon.$$

---

[3] A value $a \in \text{Im } f$ is called a regular value of $f$ if $\forall q \in \text{dom } f : f(q) = a$, it holds $\nabla f(q) \neq \mathbf{0}$.

[4] A function is proper if inverse images of compact subsets are compact.

*Proof.* Let us define the following sets

$$\begin{array}{llll}
\forall i \in [n] & : A_i & = & \{ \quad \boldsymbol{\theta}_i \in \mathbb{R}^{n_i} \quad | \quad f_i(\boldsymbol{\theta}_i) \in [p_i - \xi, p_i + \xi]\} \\
\forall j \in [m] & : B_j & = & \{ \quad \boldsymbol{\phi}_j \in \mathbb{R}^{m_j} \quad | \quad g_j(\boldsymbol{\phi}_j) \in [q_j - \xi, q_j + \xi]\}
\end{array}$$

Since $f_i$ and $g_j$ are proper $A_i$ and $B_j$ are compact sets. Thus, the continuous functions $\|\nabla f_i(\boldsymbol{\theta}_i)\|^2$ and $\|\nabla g_j(\boldsymbol{\phi}_j)\|^2$ have a minimum and maximum value on $A_i$ and $B_j$ respectively. Let us call $K_{f_i}$ and $K_{g_j}$ the maxima and $\kappa_{f_i}$ and $\kappa_{g_j}$ the minima. Observe that the minima and maxima must be all greater than zero since $[p_i - \xi, p_i + \xi]$ and $[q_j - \xi, q_j + \xi]$ are regular values. Let us define

$$\kappa = \min\{\min_{1 \le i \le n} \kappa_{f_i}, \min_{1 \le j \le m} \kappa_{g_j}\}$$

$$K = \max\{\max_{1 \le i \le n} K_{f_i}, \max_{1 \le j \le m} K_{g_j}\}$$

where $K \ge \kappa > 0$ as we discussed. Let us create the following set

$$S = \{(\boldsymbol{\theta}, \boldsymbol{\phi}) \in \mathbb{R}^N \times \mathbb{R}^M \quad | \quad \forall i \in [n] : \boldsymbol{\theta}_i \in A_i, \quad \forall j \in [m] : \boldsymbol{\phi}_j \in B_j\}$$

We can prove that every $(\boldsymbol{\theta}, \boldsymbol{\phi}) \in S$ is a safe initialization for $(\mathbf{p}, \mathbf{q})$. Of course, every $\boldsymbol{\theta}_i$ and $\boldsymbol{\phi}_j$ are not stationary points of $f_i$ and $g_j$ respectively. We also need to prove that the equilibrium $(\mathbf{p}, \mathbf{q})$ is feasible. We will prove this by contradiction. Let there be a $(\boldsymbol{\theta}, \boldsymbol{\phi}) \in S$ such that $(\mathbf{p}, \mathbf{q})$ is not feasible. Without loss of generality we can assume that there is an $i \in [n]$ such that $p_i \notin \text{Im}_{f_i}(\boldsymbol{\theta}_i)$. The case for the $g_j$ is symmetrical. Along the gradient ascent trajectory of $f_i$ with initialization at $\boldsymbol{\theta}_i$, observe that $f_i(t)$ cannot attain an infimum or a supremum in $[p_i - \xi, p_i + \xi]$ because there are no stationary points of $f_i$ in $A_i$. Observe also that at initialization $f_i(\boldsymbol{\theta}_i) \in [p_i - \xi, p_i + \xi]$. Thus $[p_i - \xi, p_i + \xi] \subseteq \text{Im}_{f_i}(\boldsymbol{\theta}_i)$, a contradiction.

Let us pick an initialization $(\boldsymbol{\theta}(0), \boldsymbol{\phi}(0))$ such that $r(0) \le \xi^2$. It is clear that $(\boldsymbol{\theta}(0), \boldsymbol{\phi}(0)) \in S$ and so it is safe for $(\mathbf{p}, \mathbf{q})$. We can do the same steps as in Theorem 2 to prove that the function $H(\mathbf{F}, \mathbf{G})$ below does not increase under the dynamics of Equation (1):

$$H(\mathbf{F}, \mathbf{G}) = \sum_{i=1}^N \int_{p_i}^{f_i} \frac{z - p_i}{\|\nabla f_i(X_{\boldsymbol{\theta}_i(0)}(z))\|^2} \mathrm{d}z + \sum_{j=1}^M \int_{q_j}^{g_j} \frac{z - q_j}{\|\nabla g_j(X_{\boldsymbol{\phi}_j(0)}(z))\|^2} \mathrm{d}z$$

Observe that since $(\boldsymbol{\theta}(0), \boldsymbol{\phi}(0)) \in S$ we have that the interval between $p_i$ and $f_i(\boldsymbol{\theta}_i(0))$ belongs in $[p_i - \xi, p_i + \xi]$ and $\|\nabla f_i(\cdot)\|^2 \ge \kappa$ in this interval. Thus we can write

$$\frac{(f_i(\boldsymbol{\theta}_i(0)) - p_i)^2}{2\kappa} \ge \int_{p_i}^{f_i(\boldsymbol{\theta}_i(0))} \frac{z - p_i}{\|\nabla f_i(X_{\boldsymbol{\theta}_i(0)}(z))\|^2} \mathrm{d}z$$

Repeating the same argument for all $f_i$ and $g_j$ we have that

$$\frac{r(0)}{2\kappa} \ge H(\mathbf{F}(\boldsymbol{\theta}(0)), \mathbf{G}(\boldsymbol{\phi}(0))) \ge H(\mathbf{F}(\boldsymbol{\theta}(t)), \mathbf{G}(\boldsymbol{\phi}(t)))$$

Let us pick $r(0) < \min\{\xi^2, \xi^2 \frac{\kappa}{K}\} = \xi^2 \frac{\kappa}{K}$. We already know that trajectories start in $S$. We will prove that they also remain in $S$. We will do this by contradiction. If a trajectory escaped $S$, then without loss of generality this means that there is at least an $i \in [n]$ such that at some $t > 0$, $f_i(\boldsymbol{\theta}_i(t)) \notin [p_i - \xi, p_i + \xi]$. The case of $g_j$ is similar. Clearly we have that

$$\int_{p_i}^{f_i(\boldsymbol{\theta}_i(t))} \frac{z - p_i}{\|\nabla f_i(X_{\boldsymbol{\theta}_i(0)}(z))\|^2} \mathrm{d}z \ge \min\left\{ \int_{p_i}^{p_i - \xi} \frac{z - p_i}{\|\nabla f_i(X_{\boldsymbol{\theta}_i(0)}(z))\|^2} \mathrm{d}z, \int_{p_i}^{p_i + \xi} \frac{z - p_i}{\|\nabla f_i(X_{\boldsymbol{\theta}_i(0)}(z))\|^2} \mathrm{d}z \right\}$$

As above, we have that the gradients in the integrals of the right hand side are less or equal than $K$ so

$$\int_{p_i}^{f_i(\boldsymbol{\theta}_i(t))} \frac{z - p_i}{\|\nabla f_i(X_{\boldsymbol{\theta}_i(0)}(z))\|^2} \mathrm{d}z \ge \frac{\xi^2}{2K}.$$

The terms of $H$ are all non-negative so we have that

$$\frac{r(0)}{2\kappa} \ge H(\mathbf{F}(\boldsymbol{\theta}(t)), \mathbf{G}(\boldsymbol{\phi}(t))) \ge \int_{p_i}^{f_i(\boldsymbol{\theta}_i(t))} \frac{z - p_i}{\|\nabla f_i(X_{\boldsymbol{\theta}_i(0)}(z))\|^2} \mathrm{d}z \ge \frac{\xi^2}{2K}.$$

But $r(0) < \xi^2 \frac{\kappa}{K}$, a contradiction. So the trajectories will stay in $S$. We can then write

$$\int_{p_i}^{f_i(\boldsymbol{\theta}_i(t))} \frac{z - p_i}{\|\nabla f_i(X_{\boldsymbol{\theta}_i(0)}(z))\|^2} \mathrm{d}z \geq \frac{(f_i(\boldsymbol{\theta}_i(t)) - p_i)^2}{2K}.$$

Repeating the same argument for all $f_i$ and $g_j$ we have that

$$\frac{r(0)}{2\kappa} \geq H(\mathbf{F}(\boldsymbol{\theta}(t)), \mathbf{G}(\boldsymbol{\phi}(t))) \geq \frac{r(t)}{2K}.$$

For every $\epsilon > 0$, there is a positive $\delta = \frac{\min\{\xi^2, \epsilon\}\kappa}{K}$ such that

$$r(0) < \delta \implies r(t) < \epsilon.$$

$\square$

A special case of the above result is the standard convex-concave games:

**Corollary 3.** *Let $L(\mathbf{x}, \mathbf{y})$ be strictly convex concave and Solution$(L)$ is the non empty set of equilibria of $L$. Then Solution$(L)$ is locally asymptotically stable for continuous GDA dynamics.*

*Proof.* The proof of the above classical result can be derived by the straightforward application of Lemma 3 for the case of $\mathbf{F}(\mathbf{x}) = \mathbf{x}$ and $\mathbf{G}(\mathbf{y}) = \mathbf{y}$. Notice that *i)* if $\mathbf{F}, \mathbf{G}$ are the identity maps all the initial configurations are safe and *ii)* if $\|\nabla \mathbf{F}\|^2 = \|\nabla \mathbf{G}\|^2 = 1$, then the initialization-dependent Lyapunov functions coincide to a single Lyapunov function, which is actually the squared Euclidean distance $r(\boldsymbol{\theta}, \boldsymbol{\phi}) = \|\mathbf{F}(\boldsymbol{\theta}) - \mathbf{p}\|^2 + \|\mathbf{G}(\boldsymbol{\phi}) - \mathbf{q}\|^2 = \|\boldsymbol{\theta} - \mathbf{p}\|^2 + \|\boldsymbol{\phi} - \mathbf{q}\|^2$. $\square$

## 8.2 HIDDEN STRICTLY CONVEX CONCAVE GAMES

### 8.2.1 GRADIENT DESCENT-ASCENT DYNAMICS

In the following preliminary result, we show that strict convexity or concavity in $L(\cdot, \cdot)$, for at least one of its arguments, suffices to yield locally asymptotic stability starting from a safe initial condition. Our argumentation leverages the power of Theorem 9 and combines the previous section stability results. Here, we will firstly outline the basic steps below:

1. We start by showing that there exists a compact set $\Omega \subset D$.

2. Therefore, since $\dot{H} \leq 0$ (Lyapunov property), any configuration $(\mathbf{F}(0), \mathbf{G}(0))$ starting from a bounded sub-level set $\Omega$ of $H$, will remain inside $\Omega$ over all time.

3. The second crucial observation is that thanks to the strictness on convexity or concavity of $L$, the largest invariant set of $\dot{H} = 0$ contains only points belonging to Von Neumann's Solution$(L)$.

Then Theorem 9 implies the local asymptotic stability of set $Z$ for Equation (3).

**Lemma 3.** *Let $L$ be strictly convex concave and $Z \subset$ Solution$(L)$ is the non empty set of equilbria of $L$ for which $(\boldsymbol{\theta}(0), \boldsymbol{\phi}(0))$ is safe. Then $Z$ is locally asymptotically stable for Equation (3).*

*Proof.* Pick a point $(\mathbf{p}, \mathbf{q}) \in Z$. Since our initialization is safe for this saddle point, we can construct the $H$ function as in Theorem 2 and prove that it has the following property

$$\dot{H} \leq 0 \text{ in } D = \{\mathrm{Im}_{f_i}(\boldsymbol{\theta}_i(0))\}_{i=1}^N \times \{\mathrm{Im}_{g_j}(\boldsymbol{\phi}_j(0))\}_{j=1}^M$$

If $(\mathbf{F}(\boldsymbol{\theta}(0)), \mathbf{G}(\boldsymbol{\phi}(0))) = (\mathbf{p}, \mathbf{q})$ then the theorem holds trivially. Otherwise, take a ball $B$ centered at the equilibirum with a small enough radius such that it is contained in the interior of $D$.

$$H_0 = \min_{(\mathbf{F}, \mathbf{G}) \in \partial B} H(\mathbf{F}, \mathbf{G})$$
$$\Omega = \{(\mathbf{F}, \mathbf{G}) \in B | H(\mathbf{F}, \mathbf{G}) \leq H_0/2\}$$

We know that in both of the cases $H_0 > 0$ from Theorem 2.

Since $\dot{H} \leq 0$, starting in $\Omega$, it implies that $H(\mathbf{F}(t), \mathbf{G}(t)) \leq H_0$ for $t \geq 0$, so $\Omega$ is forward invariant. Since $\Omega \subset D$ we know that it is bounded. $\Omega$ is closed since it is a sublevel set of a continuous function. Notice that the restriction of $\Omega$ on $B$ does not affect the above properties since $\Omega$ is in the interior of $B$. Thus $\Omega$ is a compact forward invariant set, satisfying the requirement of Theorem 9

Let $E = \{(\mathbf{F}, \mathbf{G}) \in B | \dot{H}(\mathbf{F}, \mathbf{G}) = 0\}$. Without loss of generality we can assume that $L(\cdot, \mathbf{q})$ is strictly convex as the case of $L(\mathbf{p}, \cdot)$ being strictly concave is similar. In the following inequality

$$\dot{H} \leq L(\mathbf{p}, \mathbf{G}) - L(\mathbf{p}, \mathbf{q}) + L(\mathbf{p}, \mathbf{q}) - L(\mathbf{F}, \mathbf{q}) \leq 0$$

we know that $L(\mathbf{p}, \mathbf{G}) - L(\mathbf{p}, \mathbf{q}) \leq 0$ and $L(\mathbf{p}, \mathbf{q}) - L(\mathbf{F}, \mathbf{q}) \leq 0$.

So $\dot{H} = 0$ implies $L(\mathbf{p}, \mathbf{G}) = L(\mathbf{p}, \mathbf{q}) = L(\mathbf{F}, \mathbf{q})$. By the strict convexity of $L(\cdot, \mathbf{q})$ we know that this means that $\mathbf{F} = \mathbf{p}$. Let $\mathcal{M}$ be the largest invariant set inside $E$. By the properties of $\mathcal{M}$ being invariant subset of $E$ we have

$$(\mathbf{F}(0), \mathbf{G}(0)) \in \mathcal{M} \implies \forall t : \mathbf{F}(t) = \mathbf{p} \text{ and } L(\mathbf{p}, \mathbf{G}(t)) = L(\mathbf{p}, \mathbf{q})$$

Taking the time derivatives on each of the constant quantities, they should be zero.

$$\dot{f}_i = 0 \Rightarrow \qquad \forall i \in [N]: \quad \|\nabla_{\boldsymbol{\theta}_i} f_i(X_{\boldsymbol{\theta}_i(0)}(p_i))\|^2 \frac{\partial L}{\partial f_i}(\mathbf{p}, \mathbf{G}) = 0$$

$$L(\mathbf{p}, \dot{\mathbf{G}}(t)) = 0 \Rightarrow \qquad \sum_{j=1}^{M} \|\nabla_{\boldsymbol{\phi}_j} g_j(X_{\boldsymbol{\phi}_j(0)}(g_j))\|^2 \left[\frac{\partial L}{\partial g_j}(\mathbf{p}, \mathbf{G})\right]^2 = 0$$

We know that $\|\nabla_{\boldsymbol{\theta}_i} f_i(X_{\boldsymbol{\theta}_i(0)}(p_i))\| \neq 0$ by the safety conditions and that $\|\nabla_{\boldsymbol{\phi}_j} g_j(X_{\boldsymbol{\phi}_j(0)}(g_j))\|^2 \neq 0$ inside $D$ again by safety conditions. This implies

$$\forall i \in [N] : \frac{\partial L}{\partial f_i}(\mathbf{p}, \mathbf{G}) = 0$$

$$\forall j \in [M] : \frac{\partial L}{\partial g_j}(\mathbf{p}, \mathbf{G}) = 0$$

Thus $\mathcal{M}$ contains only stationary points of $L$ so $\mathcal{M} \subseteq \text{Solution}(L)$. In addition $\mathcal{M} \subseteq D$ so only stationary points of $L$ for which the initialization is safe are allowed so $\mathcal{M} \subseteq Z$. Applying Theorem 9 we have that for any initialization of Equation (3) inside $\Omega$, as $t \to \infty$ $(\mathbf{F}(t), \mathbf{G}(t))$ approaches $\mathcal{M}$ and thus $Z$ is locally asymptotically stable for Equation (3). $\qquad \square$

A special case of the above result is the standard convex-concave games:

**Corollary 4.** *Let $L(\mathbf{x}, \mathbf{y})$ be strictly convex concave and Solution$(L)$ is the non empty set of equilibria of L. Then Solution$(L)$ is locally asymptotically stable for continuous GDA dynamics.*

In the following main result of our work, we show that strict convexity or concavity in $L(\cdot, \cdot)$, for at least one of its arguments, suffices to yield a convergence result to a Von Neumann's Solution$(L)$ starting from a safe initial condition. In order to get convergence results for any safe initialization, we need to study the region of attraction of the set $Z \subset \text{Solution}(L)$. We refine the estimation of the region of attraction as proposed in Lemma 3 by analyzing the behavior of the level sets of $H$. More precisely, we show that the proposed Lyapunov function

$$H(\mathbf{F}, \mathbf{G}) = \sum_{i=1}^{N} \int_{p_i}^{f_i} \frac{z - p_i}{\|\nabla f_i(X_{\boldsymbol{\theta}_i(0)}(z))\|^2} \mathrm{d}z + \sum_{j=1}^{M} \int_{q_j}^{g_j} \frac{z - q_j}{\|\nabla g_j(X_{\boldsymbol{\phi}_j(0)}(z))\|^2} \mathrm{d}z$$

is radially unbounded. In other words, while the operators converges to their limit values (supremum/infimum of their domain) $H \to +\infty$. In order to show that we analyze the asymptotic behavior of $\int_{c}^{\mathcal{F}} \frac{1}{\|\nabla f_i\|^2}$, while $\mathcal{F} \to \sup f_i$. Hence,

  A) Theorem 9 implies that the trajectory will approach the set of stationary points of $H$ or equivalently a set of Von Neumann's Solution$(L)$.

  B) The stability of Solution$(L)$ and Theorem 10, leads to the conclusion that the trajectory will converges to a specific point of Solution$(L)$.

**Theorem 5.** *Let $L$ be strictly convex concave and $Z \subset Solution(L)$ is the non empty set of equilibria of $L$ for which $(\boldsymbol{\theta}(0), \boldsymbol{\phi}(0))$ is safe. Under the dynamics of Equation* (1) *$(\mathbf{F}(\boldsymbol{\theta}(t)), \mathbf{G}(\boldsymbol{\theta}(t)))$ converges to a point in $Z$ as $t \to \infty$.*

*Proof.* Again let's pick a point $(\mathbf{p}, \mathbf{q}) \in Z$. Since our initialization is safe for this saddle point, we can construct the $H$ function as in Theorem 2 and prove that it has the following property

$$\dot{H} \leq 0 \text{ in } D = \{\mathrm{Im}_{f_i}(\boldsymbol{\theta}_i(0))\}_{i=1}^N \times \{\mathrm{Im}_{g_j}(\boldsymbol{\phi}_j(0))\}_{j=1}^M$$

If $(\mathbf{F}(\boldsymbol{\theta}(0)), \mathbf{G}(\boldsymbol{\phi}(0))) = (\mathbf{p}, \mathbf{q})$ then the theorem holds trivially. Otherwise define

$$H_0 = H(\mathbf{F}(\boldsymbol{\theta}(0)), \mathbf{G}(\boldsymbol{\phi}(0)))$$
$$\Omega = \{(\mathbf{F}, \mathbf{G}) \in D | H(\mathbf{F}, \mathbf{G}) \leq H_0\}$$

where we know that $H_0 > 0$ from Theorem 2. Let us assume that indeed $\Omega$ is in the interior of $D$. Then, applying the same argumentation as in Lemma 3 combined with Theorem 2, all fixed points in $Z$ are stable. So applying Theorem 10 we get that the trajectory initialized at $(\mathbf{F}(\boldsymbol{\theta}(0)), \mathbf{G}(\boldsymbol{\phi}(0))) \in \Omega$ converges to a point in $Z$. It remains to prove our assertion about the set $\Omega$:

**Claim 1.** *$\Omega$ is in the interior of $D$.*

*Proof.* We will argue that as $(\mathbf{F}, \mathbf{G})$ approaches the boundary of $D$, the value of $H$ should become unbounded. If this is true then for the finite upper bound of $H_0$, $\Omega$ should have no points close to the boundary of $H$ and thus it should be in the interior.

As $(\mathbf{F}, \mathbf{G})$ approach the boundary of $D$, at least one of the variables $f_i$ or $g_j$ approaches the endpoints points of $\mathrm{Im}_{f_i}(\boldsymbol{\theta}_i(0))$ or $\mathrm{Im}_{g_j}(\boldsymbol{\phi}_j(0))$ respectively. We will study the case of $f_i$ since the case of $g_j$ is symmetrical. The endpoint $f_{is}$ can be either the supremum or the infimum of the gradient ascent trajectory on $f_i$ or $\pm\infty$ if they do not exist. Let $f_{is}$ be the supremum or $\infty$ depending on if the former exists. We can take the gradient ascent dynamics and apply Lemma 1 to get

$$\dot{f}_i = \|\nabla_{\boldsymbol{\theta}_i} f_i(X_{\boldsymbol{\theta}_i(0)}(f_i))\|^2$$

We know that $f_i(\boldsymbol{\theta}_i(t))$ goes to $f_{is}$ when initialized at $f_i(\boldsymbol{\theta}_i(0))$. Let us define the following function

$$a(f_i) = \int_{p_i}^{f_i} \frac{1}{\|\nabla f_i(X_{\boldsymbol{\theta}_i(0)}(z))\|^2} \, dz$$

Observe that $\dot{a} = 1$, thus $\lim_{t \to \infty} a(f_i(t)) = \infty$. In other words

$$\lim_{t \to \infty} \int_{p_i}^{f_i(t)} \frac{1}{\|\nabla f_i(X_{\boldsymbol{\theta}_i(0)}(z))\|^2} \, dz = \int_{p_i}^{f_{is}} \frac{1}{\|\nabla f_i(X_{\boldsymbol{\theta}_i(0)}(z))\|^2} \, dz = \infty$$

Symmetrically if $f_{is}$ is the infimum or $-\infty$, then the limit above would be $-\infty$. In either case

$$f_i \to f_{is} \implies \int_{p_i}^{f_i} \frac{z - p_i}{\|\nabla f_i(X_{\boldsymbol{\theta}_i(0)}(z))\|^2} \, dz \to \infty$$

For the last step it is important to note that $p_i$ is not at the boundary of $D$ based on the safety conditions. Therefore as $(\mathbf{F}, \mathbf{G})$ approach the boundary of $D$ in the dynamics of Equation (3), at least one of the terms of $H$ goes to infinity. Also note that all the terms of $H$ are individually non-negative so no matter what the other variables in $(\mathbf{F}, \mathbf{G})$ are doing they cannot stop $H \to \infty$.   □

□

Again, a special case of the above result is the standard convex-concave games:

**Corollary 5.** *Let $L(\boldsymbol{x}, \boldsymbol{y})$ be strictly convex concave and $Solution(L)$ is the non empty set of equilibria of $L$. Under the continuous GDA dynamics $(\boldsymbol{x}(t), \boldsymbol{y}(t))$ converges to a point in $Solution(L)$ as $t \to \infty$.*

### 8.2.2 CONNECTIONS TO HAMILTONIAN DESCENT

In GANs numerous learning heuristics are being tested and explored. One technique that has particular interesting theoretical justification as well as practical performance is Hamiltonian Gradient Descent (HGD). Understanding the convergence guarantees for HGD is an open research question Maddison et al. (2018); Balduzzi et al. (2018); O'Donoghue & Maddison (2019). We provide some new justification about its success in GANs by provably establishing convergence of a modified version of HGD in a relatively simple but illustrative subclass of hidden convex concave games, namely 2x2 hidden bi-linear games. This class of games is fairly expressive. Despite the restriction of planar bi-linear competition in the output space, the hidden game can have an arbitrary number of variables in the parameter space. It's important to note that given the bi-linear nature of competition, the classical GDA dynamics cycles instead of converging to the equilibrium as shown in Vlatakis-Gkaragkounis et al. (2019)

More precisely, in the hidden 2x2 bi-linear game presented in Vlatakis-Gkaragkounis et al. (2019), we have two functions $f : \mathbb{R}^N \to [0, 1]$ and $g : \mathbb{R}^M \to [0, 1]$ and two constants $(p, q) \in (0, 1)^2$ where $(p, q)$ is the fully mixed equilibrium of the bi-linear game. Without loss of generality, we are interested in solving the following problem

$$\min_{\boldsymbol{\theta} \in \mathbb{R}^M} \max_{\boldsymbol{\phi} \in \mathbb{R}^N} (f(\boldsymbol{\theta}) - p)(g(\boldsymbol{\phi}) - q)$$

Defining $L(\boldsymbol{\theta}, \boldsymbol{\phi}) = (f(\boldsymbol{\theta}) - p)(g(\boldsymbol{\phi}) - q)$, the dynamics of HGD are:

$$\dot{\boldsymbol{\theta}} = -\frac{1}{2}\nabla_{\boldsymbol{\theta}}\|\nabla_{\boldsymbol{\phi}}L(\boldsymbol{\theta}, \boldsymbol{\phi})\|^2 - \frac{1}{2}\nabla_{\boldsymbol{\theta}}\|\nabla_{\boldsymbol{\theta}}L(\boldsymbol{\theta}, \boldsymbol{\phi})\|^2$$

$$\dot{\boldsymbol{\phi}} = -\frac{1}{2}\nabla_{\boldsymbol{\phi}}\|\nabla_{\boldsymbol{\theta}}L(\boldsymbol{\theta}, \boldsymbol{\phi})\|^2 - \frac{1}{2}\nabla_{\boldsymbol{\phi}}\|\nabla_{\boldsymbol{\phi}}L(\boldsymbol{\theta}, \boldsymbol{\phi})\|^2$$

(6)

Observe that the second term of each right hand side would be zero in a classical bi-linear game but involves second order derivatives of $f$ and $g$ in the case of hidden bi-linear games. To circumvent the complexities of the second order derivatives and mimic the classical bi-linear game we will study a modified version of Equation (6), namely:

$$\dot{\boldsymbol{\theta}} = -\frac{1}{2}\nabla_{\boldsymbol{\theta}}\|\nabla_{\boldsymbol{\phi}}L(\boldsymbol{\theta}, \boldsymbol{\phi})\|^2 \qquad \dot{\boldsymbol{\phi}} = -\frac{1}{2}\nabla_{\boldsymbol{\phi}}\|\nabla_{\boldsymbol{\theta}}L(\boldsymbol{\theta}, \boldsymbol{\phi})\|^2$$

(7)

Employing an analysis similar to the one in Section 3.2, we get the following convergence result:

**Theorem 13.** *Let $(\boldsymbol{\theta}(0), \boldsymbol{\phi}(0))$ be safe for $(p, q)$. Then $(f(\boldsymbol{\theta}(t)), g(\boldsymbol{\phi}(t)))$ converges to $(p, q)$ under the dynamics of Equation* (7).

*Proof.* Simple substitution gives us

$$\dot{\boldsymbol{\theta}} = -\nabla_{\boldsymbol{\theta}}f(\boldsymbol{\theta})\|\nabla_{\boldsymbol{\phi}}g(\boldsymbol{\phi})\|^2(f(\boldsymbol{\theta}) - p)$$

$$\dot{\boldsymbol{\phi}} = -\nabla_{\boldsymbol{\phi}}g(\boldsymbol{\phi})\|\nabla_{\boldsymbol{\theta}}f(\boldsymbol{\theta})\|^2(g(\boldsymbol{\phi}) - q)$$

Applying Lemma 1 and following the same steps as before

$$\dot{f} = -\|\nabla_{\boldsymbol{\theta}}f(X_{\boldsymbol{\theta}(0)}(f))\|^2\|\nabla_{\boldsymbol{\phi}}g(X_{\boldsymbol{\phi}(0)}(g))\|^2(f - p)$$

$$\dot{g} = -\|\nabla_{\boldsymbol{\phi}}g(X_{\boldsymbol{\phi}(0)}(g))\|^2\|\nabla_{\boldsymbol{\theta}}f(X_{\boldsymbol{\theta}(0)}(f))\|^2(g - q)$$

Once again we consider the function

$$H(f, g) = \int_p^f \frac{z - p}{\|\nabla f(X_{\boldsymbol{\theta}(0)}(z))\|^2}\mathrm{d}z + \int_q^g \frac{z - q}{\|\nabla g(X_{\boldsymbol{\phi}(0)}(z))\|^2}\mathrm{d}z$$

Simple substitution gives

$$\dot{H} = -(f - p)\left(\|\nabla_{\boldsymbol{\phi}}g(X_{\boldsymbol{\phi}(0)}(g))\|^2(f - p)\right) - (g - q)\left(\|\nabla_{\boldsymbol{\theta}}f(X_{\boldsymbol{\theta}(0)}(f))\|^2(g - q)\right)$$

A little bit of reorganization gives

$$\dot{H} = -(f - p)^2\|\nabla_{\boldsymbol{\phi}}g(X_{\boldsymbol{\phi}(0)}(g))\|^2 - (g - q)^2\|\nabla_{\boldsymbol{\theta}}f(X_{\boldsymbol{\theta}(0)}(f))\|^2 \leq 0$$

Thus, we get

$$\dot{H} \leq 0 \text{ in } D = \text{Im}_f(\boldsymbol{\theta}(0)) \times \text{Im}_g(\boldsymbol{\phi}(0))$$

Similarly with the strict convex analysis of the previous section, if $(f(\boldsymbol{\theta}(0)), g(\boldsymbol{\phi}(0))) = (p, q)$ then the theorem holds trivially. Otherwise define

$$H_0 = H(f(\boldsymbol{\theta}(0)), g(\boldsymbol{\phi}(0)))$$
$$\Omega = \{(f, g) \in D | H(f, g) \leq H_0\}$$

where we know that $H_0 > 0$ from Theorem 2. Additionally, we can apply Claim 1 even in the new dynamics, so $\Omega$ is in the interior of $D$. Since $\dot{H} \leq 0$, starting in $\Omega$, it implies that $H(f(t), g(t)) \leq H_0$ for $t \geq 0$, so $f(t), g(t)$ stays in $\Omega$. Additionally, $\Omega$ is closed since it is a sublevel set of a continuous function. Notice that the restriction of $\Omega$ on $D$ does not affect the above properties since $\Omega$ is in the interior of $D$. Thus $\Omega$ is a compact forward invariant set.

For a safe initialization $(\boldsymbol{\theta}(0), \boldsymbol{\phi}(0))$, both $\|\nabla_{\boldsymbol{\phi}} g(X_{\boldsymbol{\phi}(0)}(g(t)))\|, \|\nabla_{\boldsymbol{\theta}} f(X_{\boldsymbol{\theta}(0)}(f(t)))\|$ cannot go to 0 as this happens only at the boundaries of $D$ which are outside $\Omega$. So $\dot{H} = 0$ only at $(p, q)$ in $\Omega$.

Therefore, applying Theorem 9, we get that $(f(\boldsymbol{\theta}(t)), g(\boldsymbol{\phi}(t)))$ converges to $(p, q)$

$\square$

## 8.3  REGULARIZATION AND CONVERGENCE

> In this section, we show that even in the absence of strict convexity/concavity for both of the operators, it is possible to achieve a positive convergence result by sacrificing the exactness of a targeted equilibrium. In other words, we prove that by adding a small regularization term, the new utility function becomes strictly convex strictly concave. Beside the guaranteed convergence of the "perturbed" $L'$, we can always choose sufficiently small magnitude of regularization such that the new equilibria are arbitrarily close to the initial ones.

**Theorem 6.** *If $L$ is a convex concave function with invertible Hessians at all its equilibria, then for each $\epsilon > 0$ there is a $\lambda > 0$ such that $L'$ has equilibria that are $\epsilon$-close to the ones of $L$.*

*Proof.* For any choice of $\lambda > 0$ we have that $L'$ is strictly convex strictly concave so the KKT conditions are sufficient to determine its equilibria.

$$\frac{\partial L(\mathbf{x}, \mathbf{y})}{\partial x_i} + \lambda x_i = 0$$
$$\frac{\partial L(\mathbf{x}, \mathbf{y})}{\partial y_j} - \lambda y_j = 0$$

We can view the above set of constraints as a single vector constraint $r(\lambda, \mathbf{x}, \mathbf{y}) = \mathbf{0}$. Note that by assumption of the Hessians being invertible at all equilibria, $L$ has a unique equilibrium $(\mathbf{x}^*, \mathbf{y}^*)$. Clearly we have that $r(0, \mathbf{x}^*, \mathbf{y}^*) = \mathbf{0}$. Observe that for the Jacobian of $r$ at $(0, \mathbf{x}^*, \mathbf{y}^*)$ with respect to $(\mathbf{x}, \mathbf{y})$ we have that

$$D_{(\mathbf{x}, \mathbf{y})} r(0, \mathbf{x}^*, \mathbf{y}^*) = \nabla^2 L(\mathbf{x}^*, \mathbf{y}^*)$$

and thus it is invertible. Invoking the Implicit function Theorem, there is a differentiable function $g$, defined in a small enough neighborhood of 0, that takes a $\lambda$ and returns $g(\lambda) = (\mathbf{x}(\lambda), \mathbf{y}(\lambda))$ such that $r(\lambda, g(\lambda)) = \mathbf{0}$. Thus for a small enough $\lambda$, we have that $g$ returns the corresponding equilibria of $L'$. By continuity of $g$, for all $\epsilon$ there is a $\delta > 0$

$$\forall 0 < \lambda < \delta : \|\mathbf{x}(\lambda) - \mathbf{x}(0)\|^2 + \|\mathbf{y}(\lambda) - \mathbf{y}(0)\|^2 \leq \epsilon^2$$

But $(\mathbf{x}(0), \mathbf{y}(0)) = (\mathbf{x}^*, \mathbf{y}^*)$ so the equilbrium of $L'$ has an $\epsilon$-close equilibrium of $L$ for $\lambda < \delta$. By strict convexity strict concavity of $L'$, it has a unique equilibrium as well. So the equilibria of $L'$ and $L$ are $\epsilon$-close to each other. $\square$

> The previous theorem highlights that small values of $\lambda$ induce only small changes to the equilibria of the hidden game. As is the case for classical convex concave games, larger values of $\lambda$ lead to (exponentially) faster convergence. To prove this for HCC games, we provide a detailed upper and lower bound analysis of the gradients of $f_i$ and $g_j$.

**Theorem 7.** *Let $(\boldsymbol{\theta}(0), \boldsymbol{\phi}(0))$ be a safe initialization for the unique equilibrium of $L'(\mathbf{p}, \mathbf{q})$. If*
$$r(t) = \|\mathbf{F}(\boldsymbol{\theta}(t)) - \mathbf{p}\|^2 + \|\mathbf{G}(\boldsymbol{\phi}(t)) - \mathbf{q}\|^2$$
*then there are initialization dependent constants $c_0, c_1 > 0$ such that $r(t) \leq c_0 \exp(-\lambda c_1 t)$.*

*Proof.* Following the same analysis with the strict convex concave analysis of the previous section, if $(\mathbf{F}(\boldsymbol{\theta}(0)), \mathbf{G}(\boldsymbol{\phi}(0))) = (\mathbf{p}, \mathbf{q})$ then the theorem holds trivially. Otherwise, since our initialization is safe for $(\mathbf{p}, \mathbf{q})$, we can construct the $H$ function as in Theorem 2 and prove that it has the following property in $D = \{\text{Im}_{f_i}(\boldsymbol{\theta}_i(0))\}_{i=1}^N \times \{\text{Im}_{g_j}(\boldsymbol{\phi}_j(0))\}_{j=1}^M$

$$\dot{H} \leq L'(\mathbf{p}, \mathbf{G}) - L'(\mathbf{p}, \mathbf{q}) + L'(\mathbf{p}, \mathbf{q}) - L'(\mathbf{F}, \mathbf{q})$$

$$\leq -\frac{\lambda}{2} \left( \|\mathbf{F}(\boldsymbol{\theta}(t)) - \mathbf{p}\|^2 + \|\mathbf{G}(\boldsymbol{\phi}(t)) - \mathbf{q}\|^2 \right)$$

$$\leq -\frac{\lambda}{2} r(t)$$

Where the second step follows from $L'(\mathbf{p}, \cdot)$ being $\lambda$ strongly concave and $L'(\cdot, \mathbf{q})$ being $\lambda$ strongly convex and $\mathbf{q}, \mathbf{p}$ being the corresponding optima of these functions since $(\mathbf{p}, \mathbf{q})$ is an equilibrium. Let us define

$$H_0 = H(\mathbf{F}(\boldsymbol{\theta}(0)), \mathbf{G}(\boldsymbol{\phi}(0)))$$
$$\Omega = \{(\mathbf{F}, \mathbf{G}) \in D | H(\mathbf{F}, \mathbf{G}) \leq H_0\}$$

where we know that $H_0 > 0$ from Theorem 2. Additionally, we can apply Claim 1 even in the new dynamics, so $\Omega$ is in the interior of $D$. Since $\dot{H} \leq 0$, starting in $\Omega$, it implies that $H(\mathbf{F}(\boldsymbol{\theta}(t)), \mathbf{G}(\boldsymbol{\phi}(t))) \leq H_0$ for $t \geq 0$, so $(\mathbf{F}(t), \mathbf{G}(t))$ stays in $\Omega$. Additionally, $\Omega$ is closed since it is a sublevel set of a continuous function. Notice that the restriction of $\Omega$ on $D$ does not affect the above properties since $\Omega$ is in the interior of $D$. Thus $\Omega$ is a compact forward invariant set.

For a safe initialization $(\boldsymbol{\theta}(0), \boldsymbol{\phi}(0))$, the following continuous functions must have a minimum and maximum value on $\Omega$ respectively.

$$K_{f_i} \geq \|\nabla f_i(X_{\boldsymbol{\theta}_i(0)}(\cdot))\|^2 \geq \kappa_{f_i}$$
$$K_{g_j} \geq \|\nabla g_j(X_{\boldsymbol{\phi}_j(0)}(\cdot))\|^2 \geq \kappa_{g_j}$$

Observe that the minima and maxima must be all greater than zero , since both $\|\nabla_{\boldsymbol{\phi}_j} g_j(X_{\boldsymbol{\phi}_j(0)}(g(t)))\|, \|\nabla_{\boldsymbol{\theta}_i} f_i(X_{\boldsymbol{\theta}_i(0)}(f(t)))\|$ cannot go to 0 as this happens only at the boundaries of $D$ which are outside $\Omega$.

Let us define

$$\kappa = \min\{\min_{1 \leq i \leq n} \kappa_{f_i}, \min_{1 \leq j \leq m} \kappa_{g_j}\}$$
$$K = \max\{\max_{1 \leq i \leq n} K_{f_i}, \max_{1 \leq j \leq m} K_{g_j}\}$$

Observe that $K \geq \|\nabla f_i(X_{\boldsymbol{\theta}_i(0)}(\cdot))\|^2 \geq \kappa$ in this interval. Thus we can write

$$\frac{(f_i(\boldsymbol{\theta}_i(t)) - p_i)^2}{2\kappa} \geq \int_{p_i}^{f_i(\boldsymbol{\theta}_i(t))} \frac{z - p_i}{\|\nabla f_i(X_{\boldsymbol{\theta}_i(0)}(z))\|^2} dz \geq \frac{(f_i(\boldsymbol{\theta}_i(t)) - p_i)^2}{2K}$$

Repeating the same argument for all $f_i$ and $g_j$ we have that

$$\frac{r(t)}{2\kappa} \geq H(\mathbf{F}(\boldsymbol{\theta}(t)), \mathbf{G}(\boldsymbol{\phi}(t))) \geq \frac{r(t)}{2K}$$

Thus we can extend our analysis

$$\dot{H} \leq -\lambda r(t) \leq -\frac{2\kappa\lambda}{2} H(t) \Rightarrow H(t) \leq H_0 e^{-\lambda\kappa t} \Rightarrow r(t) \leq 2 \times K \times H_0 e^{-\lambda\kappa t}$$

$\square$

# 9 APPLICATIONS

## 9.1 CONNECTING GANS AND HIDDEN CONVEX-CONCAVE GAMES

At the heart of many GAN formulations like the standard GAN Goodfellow et al. (2014b), f-GAN Nowozin et al. (2016) and Wassertein GAN (WGAN) Arjovsky et al. (2017) lies a classical convex concave game in the operator output space. Indeed for the realizeable case Goodfellow et al. (2014b) used the underlying convexity properties to find the Nash equilibria of standard GAN and Farnia & Ozdaglar (2020) did the same thing for the f-GAN and WGAN. Perhaps surprisingly, neither work references explicitly the convex concave nature of the operator output space game or von Neumann's minimax theorem. To highlight the significance of von Neumann equilibria as a solution concept for GANs, we show how the optimal $G^*$ and $D^*$ can be derived separately from each other by solving the corresponding min-max (max-min) problems. This allows one to independently verify the validity of von Neumann's minimax theorem and its generalizations for GANs. We also extend our analysis to a wide class of non-realizeable cases as well.

In practice however, as noted explicitly by Goodfellow (2017), the updates in GAN training happen in the parameter space giving rise to a HCC game. This has exactly motivated studying the learning dynamics of HCC games in Section 3.

Thus, in this section, we present these connections between Hidden Convex-Concave games and the different architectures of Generative Adversarial Networks. More specifically, we start by exploring the structure of GANs and we verify their hidden convex-concave intrinsic form.

1. *Under this scope of hidden games*, the strong (or even strict) convexity/concavity of at least one of the players (Discriminator/Generator) in combination with the convergence results of the following sections provide some theoretical explanation about the convergence properties of those architectures even under the vanilla Gradient Descent-Ascent Dynamics.

2. To indicate the relation of Von-Neumann solution with this hidden model, we leverage this hidden convex-concave structure in order to compute the well-known both $\min\max$ and $\max\min$ optima of GANs under the realizability or not assumption. The results of this section are summarized in the following table:

| Type of GAN | $G^*$ | $D^*$ | Hidden Structure |
|---|---|---|---|
| GAN | $p_{\text{data}}$ | $\frac{1}{2}$ | Linear VS Strongly-Concave |
| GAN | $\arg\min_{G\in\mathcal{G}} \text{JSD}(p_{\text{data}}||p_{\text{G}})$ | $\frac{p_{\text{data}}}{p_{\text{data}}+p_{\text{G}^*}}$ | Linear VS Strongly-Concave |
| f-GAN | $p_{\text{data}}$ | $f'(1)$ | Linear VS Concave |
| f-GAN | $\arg\min_{G\in\mathcal{G}} \text{D}_f(p_{\text{data}}||p_{\text{G}})$ | $f'\left(\frac{p_{\text{data}}}{p_{\text{G}^*}}\right)$ | Linear VS Concave |
| WGAN | $p_{\text{data}}$ | $c$ | Linear VS Linear |
| WGAN | $\arg\min_{G\in\mathcal{G}} \text{EMD}(p_{\text{data}}||p_{\text{G}})$ | – | Linear VS Linear |

Table 1: $p_{\text{data}}$ represents the target data distribution. $G^*$ is the min-max generator and $D^*$ is the max-min discriminator. JSD denotes the Jensen–Shannon divergence, $\text{D}_f$ the $f$-divergence for the convex function $f$ and EMD the earth mover distance and $c$ the constant discriminator. xGAN, xGAN correspond to the realizable and the non-realizable case accordingly. – indicates the lack of a closed form solution for $D^*$ of WGAN.

In the following three subsections, we analyze both the derivation of $\arg\min\max$ and $\arg\max\min$ for the **"vanilla-GANs", f-GANs, W-GANs** using min-max optimization arguments based on the Minimax Theorem for convex-concave functions. More precisely,

1. In the Lemmas 4, 9 and 14, we present the optimal discriminators which consist the best-response for the case of a fixed generator. In all these maximization problems, typically each $D(x)$ is decoupled and $D_G^*(x)$ is derived by the hidden concavity of the discriminator architecture.

2. In the Lemmas 5, 10 and 15, we present the optimal generators which consist the best-response for the case of a fixed discriminator. In all these minimization problems, typically the generator can cheat the fixed discriminator by producing greedily a distribution only over the restricted subset of the points for which the discriminator has the highest confidence about their originality.

3. In the Lemmas 6, 11 and 16, we leverage lemmas of (Item 1) to understand the form GAN's utility function which corresponds typically to JSD, $f$-divergence and Wasserstein distance which donate their name to their GAN architecture as well. Thus, it is then trivial to show that $p_{\text{data}}$ is the optimal choice in the realizable case.

4. In the Lemmas 7, 12 and 17, on the other side of the coin, we emphasize to derive the minmax solutions too. Our proof strategy invokes the partition to two basic sets, $S_{G_D^*}$ and $S_{G_D^*}^c$, the "preferable" or not data points by the generator. Leveraging the concavity part of the objective, we show that the best strategy for the discriminator is to label all the points uniformly with the same confidence in order to incentivize the generator to expands its support to the maximum possible.

5. In the Lemmas 8 and 13, we analyze the non-realizable case. One the one hand using Item 3 we are able to compute the $\arg\max\min$ generator $G^*$. To conclude about the $\arg\min\max$ discriminators we apply the Von Neumann's Minimax theorem to prove $D^* = \text{Best-Response}(G^*)$.

### 9.1.1 GAN

The utility of the zero-sum game $V(G, D)$ for the distribution $p_{\text{data}}$ over the discrete set $\mathcal{N}$ is

$$V(G, D) = \sum_{x \in \mathcal{N}} p_{\text{data}}(x) \log(D(x)) + \sum_{x \in \mathcal{N}} p_{\text{G}}(x) \log(1 - D(x))$$

On the one hand, it is easy to check that for a fixed discriminator $D$, the utility function is linear over the $p_{\text{G}}$ operator. On the other hand, for a fixed generator $G$, the utility function is of the form $a \log(D) + b \log(1 - D)$ which is strongly-concave.

We start our work with the following lemmas

**Lemma 4** (Goodfellow et al. (2014b)). *For a fixed generator $G$ the optimal discriminator is*

$$D_G^*(x) = \frac{p_{data}(x)}{p_{data}(x) + p_G(x)}$$

*Proof.* Observe that the optimization problem for each $D(x)$ is decoupled. Thus

$$D_G^*(x) = \arg\max_{D \in [0,1]} p_{\text{data}}(x) \log(D) + p_{\text{G}}(x) \log(1 - D)$$

By concavity the unique maximum of the above is given by

$$D_G^*(x) = \frac{p_{\text{data}}(x)}{p_{\text{data}}(x) + p_{\text{G}}(x)}$$

$\square$

**Lemma 5.** *For a fixed discriminator $D$, any distribution supported only on*

$$S_{G_D^*} = \{x \in \mathcal{N} : \forall x' \in \mathcal{N} \quad D(x) \geq D(x')\}$$

*is an optimal generator when it is allowed to choose any distribution over $\mathcal{N}$.*

*Proof.* Observe that for a fixed discriminator, the optimal generator optimizes

$$\sum_{x \in \mathcal{N}} p_{\text{G}}(x) \log(1 - D(x))$$

since the other term is independent of the generator. Let us define the following

$$D_{\max} = \max_{x \in \mathcal{N}} D(x)$$

Then we have that

$$\sum_{x \in \mathcal{N}} p_{\text{G}}(x) \log(1 - D(x)) \geq \log(1 - D_{\max})$$

with the equality being true only for distributions supported only on $S_{G_D^*}$. $\qquad\square$

**Lemma 6** (Goodfellow et al. (2014b)). *The min-max generator is the following distribution*

$$G^* = \arg\min_{G \in \mathcal{G}} \text{JSD}(p_{data} || p_G).$$

*Proof.* We can substitute in $V(G, D)$ the optimal discriminator from Lemma 4. Thus we get

$$V(G, D_G^*) = \sum_{x \in \mathcal{N}} p_{\text{data}}(x) \log\left(\frac{p_{\text{data}}(x)}{p_{\text{data}}(x) + p_{\text{G}}(x)}\right) + \sum_{x \in \mathcal{N}} p_{\text{G}}(x) \log\left(\frac{p_{\text{G}}(x)}{p_{\text{data}}(x) + p_{\text{G}}(x)}\right)$$

We can now prove that

$$V(G, D_G^*) = -\log(4) + \text{KL}\left(p_{\text{data}} || \frac{p_{\text{G}} + p_{\text{data}}}{2}\right) + \text{KL}\left(p_{\text{G}} || \frac{p_{\text{G}} + p_{\text{data}}}{2}\right)$$

$$= -\log(4) + 2\text{JSD}(p_{\text{data}} || p_{\text{G}})$$

By minimizing $V(G, D_G^*)$, the result follows trivially. $\qquad\square$

**Lemma 7.** *The max-min discriminator is*

$$\forall x \in \mathcal{N} : D^*(x) = \frac{1}{2}$$

*when the generator is allowed choose any distribution over $\mathcal{N}$,*

*Proof.* We can substitute in $V(G, D)$ the optimal generator from Lemma 5

$$V(G_D^*, D) = \log(D_{\max}) \sum_{x \in S_{G_D^*}} p_{\text{data}}(x) + \log(1 - D_{\max}) \sum_{x \in S_{G_D^*}} p_{\text{G}}(x)$$

$$+ \sum_{x \notin S_{G_D^*}} p_{\text{data}}(x) \log(D(x)) + \sum_{x \notin S_{G_D^*}} \underbrace{p_{\text{G}}(x)}_{0} \log(1 - D(x))$$

Observe that for $x \notin S_{G_D^*}$, if $D$ takes more than two values then setting $D$ equal to the highest of the them for all $x \notin S_{G_D^*}$ improves utility. So for an optimal discriminator we would have a single value $D_{\max} > D_{\min}$. In the end we have that

$$x \notin S_{G_D^*} \implies D^*(x) = D_{\min}$$

$$x \in S_{G_D^*} \implies D^*(x) = D_{\max}$$

Observe that for any combination of $D_{\max}$ and $D_{\min}$ with $D_{\max} > D_{\min}$, the constant discriminator $D_{\max}$ has higher utility. Therefore we can focus our attention on the constant discriminator $D_{\text{const}}(x) = D$

$$V(G_{D_{\text{const}}}^*, D_{\text{const}}) = \log(D) + \log(1 - D)$$

The optimal value for $D$ is $\frac{1}{2}$ and as a result

$$D^*(x) = \frac{1}{2}.$$

$\qquad\square$

**Lemma 8** (Non-realizable case). *If we assume that choice of generator $G$ is restricted in $\mathcal{G}$, a convex compact subset of the $|\mathcal{N}|$ dimensional simplex, such that $p_{data} \notin \mathcal{G}$. Then*

$$(G^*, D^*) = \left( \underset{G \in \mathcal{G}}{\arg\min} \, \text{JSD}(p_{data} || p_G), \frac{p_{data}}{p_{data} + p_{G^*}} \right) 5$$

*Proof.* We cannot readily apply von Neumann's minimax theorem since the $V(G, D)$ may be infinite at the boundary points of $\mathcal{D} = (0, 1)^{|\mathcal{N}|}$ for the discriminator. We can still apply Fan's Minimax Theorem

$$\min_{G \in \mathcal{G}} \sup_{D \in \mathcal{D}} V(G, D) = \sup_{D \in \mathcal{D}} \min_{G \in \mathcal{G}} V(G, D).$$

It is easy to check that Lemma 6 holds even in the non-realizable case. As a result, the generator is minimizing $\text{JSD}(p_{\text{data}} || p_G)$ whose value is finite. Clearly the quantities above are finite. Thus there exists a real number $v$, the value of the game, such that:

$$\begin{cases} \forall D \in \mathcal{D} & : V(G^*, D) \leq v = V(G^*, D^*) \quad (A) \\ \forall G \in \mathcal{G} & : V(G, D^*) \geq v = V(G^*, D^*) \quad (B) \end{cases}$$

for $G^*$ the minimizer of $\text{JSD}(p_{\text{data}} || p_G)$ and a $D^* \in [0, 1]^{|\mathcal{N}|}$. Now applying Lemma 4, we have that

$$\tilde{D} = \text{Best-Response}(G^*) = \frac{p_{\text{data}}}{p_{\text{data}} + p_{G^*}}.$$

Additionally, by the optimality of the response and the consequence (A) of Minimax Theorem it holds that $V(G^*, \tilde{D}) = v$. Finally, since $V(G^*, \cdot)$ is strongly concave, all other discriminators receive value less than $v$ and are not optimal. Thus

$$D^* = \tilde{D} = \frac{p_{\text{data}}}{p_{\text{data}} + p_{G^*}}$$

$\square$

### 9.1.2 F-GAN

The utility of the zero-sum game $V(G, D)$ for the distribution $p_{\text{data}}$ over the discrete set $\mathcal{N}$ is

$$V(G, D) = \sum_{x \in \mathcal{N}} p_{\text{data}}(x) D(x) - \sum_{x \in \mathcal{N}} p_G(x) f^*(D(x))$$

We will assume that $f$ is a strictly convex function with $f(1) = 0$. On the one hand, it is easy to check that for a fixed discriminator $D$, the utility function is linear over the $p_G$ operator. On the other hand, for a fixed generator $G$, the utility function is of the form $aD - bf^*(D)$ which is strictly-concave.

We start our work with the following lemmas

**Lemma 9** (Nowozin et al. (2016)). *For a fixed generator $G$ the optimal discriminator is*

$$D_G^*(x) = f' \left( \frac{p_{data}(x)}{p_G(x)} \right)$$

*Proof.* Observe that the optimization problem for each $D(x)$ is decoupled. Thus

$$D_G^*(x) = \underset{D}{\arg\max} \, p_{\text{data}}(x) D - p_G(x) f^*(D)$$

By concavity the unique maximum of the above is given by Fermat criterion

$$D_G^*(x) = ((f^*)')^{-1} \left( \frac{p_{\text{data}}(x)}{p_G(x)} \right) = f' \left( \frac{p_{\text{data}}(x)}{p_G(x)} \right)$$

$\square$

---

[5]We note that $D^*$, may take the value 1 for some $x \in \mathcal{N}$ if the generator $G^*$ does not have full support. Assigning $D(x) = 1$ for some $x$ may lead to infinite utilities in general. We prove however for that the pair $(G^*, D^*)$ this is not the case. We thus consider that pair an equilibrium.

**Lemma 10.** *For a fixed discriminator D, any distribution supported only on*

$$S_{G_D^*} = \{x \in \mathcal{N} : \forall x' \in \mathcal{N} \quad f^*(D(x)) \geq f^*(D(x'))\}$$

*is an optimal generator when it is allowed to choose any distribution over $\mathcal{N}$.*

*Proof.* Observe that for a fixed discriminator, the optimal generator optimizes

$$- \sum_{x \in \mathcal{N}} p_{\mathrm{G}}(x) f^*(D(x))$$

since the other term is independent of the generator. Let us define the following

$$F_{\max} = \max_{x \in \mathcal{N}} f^*(D(x))$$

Then we have that

$$- \sum_{x \in \mathcal{N}} p_{\mathrm{G}}(x) f^*(D(x)) \geq -F_{\max}$$

with the equality being true only for distributions supported only on $S_{G_D^*}$. $\qquad\square$

**Lemma 11** (Nowozin et al. (2016)). *The min-max generator is the following distribution*

$$G^* = \arg\min_{G \in \mathcal{G}} \mathrm{D}_f(p_{data} || p_G).$$

*Proof.* We can substitute in $V(G, D)$ the optimal discriminator from Lemma 9. Thus we get

$$V(G, D_G^*) = \sum_{x \in \mathcal{N}} p_{\mathrm{data}}(x) f'\left(\frac{p_{\mathrm{data}}(x)}{p_{\mathrm{G}}(x)}\right) - \sum_{x \in \mathcal{N}} p_{\mathrm{G}}(x) f^*\left(f'\left(\frac{p_{\mathrm{data}}(x)}{p_{\mathrm{G}}(x)}\right)\right)$$

We will first prove that:

$$V(G, D_G^*) = \mathrm{D}_f(p_{\mathrm{data}} || p_{\mathrm{G}})$$

Let's recall firstly the definition of f-divergence:

$$\mathrm{D}_f(p_{\mathrm{data}} || p_{\mathrm{G}}) = \sum_{x \in \mathcal{N}} p_{\mathrm{G}}(x) f\left(\frac{p_{\mathrm{data}}(x)}{p_{\mathrm{G}}(x)}\right)$$

Since $f$ is convex and lower semi-continuous, Frenchel convex duality guarantees that we can write $f$ in terms of its conjugate dual as $f(u) = \sup_{v \in \mathbb{R}} \{uv - f^*(v)\}$. Equivalently we get:

$$\begin{aligned} \mathrm{D}_f(p_{\mathrm{data}} || p_{\mathrm{G}}) &= \sum_{x \in \mathcal{N}} p_{\mathrm{G}}(x) \sup_{v \in \mathbb{R}} \left\{ \left(\frac{p_{\mathrm{data}}(x)}{p_{\mathrm{G}}(x)}\right) v - f^*(v) \right\} \\ &= \sum_{x \in \mathcal{N}} \sup_{v \in \mathbb{R}} \{ p_{\mathrm{data}}(x) v - f^*(v) p_{\mathrm{G}}(x) \} \\ &= \sum_{x \in \mathcal{N}} p_{\mathrm{data}}(x) f'\left(\frac{p_{\mathrm{data}}(x)}{p_{\mathrm{G}}(x)}\right) - \sum_{x \in \mathcal{N}} p_{\mathrm{G}}(x) f^*\left(f'\left(\frac{p_{\mathrm{data}}(x)}{p_{\mathrm{G}}(x)}\right)\right) \end{aligned}$$

The last line follows arguments similar to Lemma 9 applied for each term. By minimizing $V(G, D_G^*)$, the result follows trivially. $\qquad\square$

**Lemma 12.** *The max-min discriminator is*

$$\forall x \in \mathcal{N} : D^*(x) = f'(1)$$

*when the generator is allowed choose any distribution over $\mathcal{N}$.*

*Proof.* We want to substitute in $V(G, D)$ the optimal generator from Lemma 5. Observe that for all $x \in S_{G_D^*}$, we may not have all $D(x)$ to be equal. Only the values of $f^*$ are guaranteed to be equal, $f^*(D(x)) = F_{\max}$. However, if there are two distinct $D$ values then we can always pick the higher one and improve utility. Thus we can focus on discriminators that are constant over $S_{G_D^*}$. Let $D_{F_{\max}}$ be the corresponding value

$$V(G_D^*, D) = D_{F_{\max}} \sum_{x \in S_i} p_{\text{data}}(x) - f^*(D_{F_{\max}}) \sum_{x \in S_i} p_{\text{G}}(x)$$
$$+ \sum_{x \notin S_{G_D^*}} p_{\text{data}}(x)D(x) - \sum_{x \notin S_{G_D^*}} \underbrace{p_{\text{G}}(x)}_{0} f^*(D(x))$$

Observe that for $x \notin S_{G_D^*}$, if $D$ takes more than two values then setting $D$ equal to the highest of the them for all $x \notin S_{G_D^*}$ improves utility. So for an optimal discriminator we would have a single value $D_{F_{\min}}$ with $f^*(D_{F_{\min}}) < f^*(D_{F_{\max}})$. As a result

$$x \notin S_{G_D^*} \implies D^*(x) = D_{F_{\min}}$$
$$x \in S_{G_D^*} \implies D^*(x) = D_{F_{\max}}$$

We now have two cases. For any combination with $D_{F_{\min}} > D_{F_{\max}}$, the constant discriminator $D(x) = D_{F_{\min}}$ has higher utility. Symmetrically, for any combination with $D_{F_{\max}} > D_{F_{\min}}$, the constant discriminator $D(x) = D_{F_{\max}}$ has higher utility. Thus the optimal discriminator is constant. Plugging in the constant discriminator $D_{\text{const}}(x) = D$ we get

$$V(G_{D_{\text{const}}}^*, D_{\text{const}}) = D + f^*(D)$$

The optimal value for $D$ follwoing the approach of Lemma 9 is $f'(1)$ and as a result

$$D^*(x) = f'(1)$$

$\square$

**Lemma 13** (Non-realizable case). *Assume that $f \in C^1$ is strictly convex and $\lim_{x \to 0^+} x f(\frac{1}{x})$ exists and is finite[6]. If the choice of generator $G$ is restricted in $\mathcal{G}$, a convex compact subset of the $|\mathcal{N}|$ dimensional simplex, such that $p_{data} \notin \mathcal{G}$ then*

$$(G^*, D^*) = \left( \underset{G \in \mathcal{G}}{\arg \min} \, D_f(p_{data} || p_G), f'\left( \frac{p_{data}}{p_{G^*}} \right) \right)$$

*Proof.* We cannot readily apply von Neumann's minimax theorem since the $V(G, D)$ since $\mathcal{D} = \mathbb{R}^{|\mathcal{N}|}$ is not compact for the discriminator. We can still apply Fan's Minimax Theorem

$$\min_{G \in \mathcal{G}} \sup_{D \in \mathcal{D}} V(G, D) = \sup_{D \in \mathcal{D}} \min_{G \in \mathcal{G}} V(G, D).$$

It is easy to check that Lemma 12 holds even in the non-realizable case. As a result, the generator is minimizing $D_f(p_{\text{data}} || p_G)$ whose value is finite under the assumptions we made on $f$. Clearly the quantities above are finite. Thus there exists a real number $v$, the value of the game, such that:

$$\begin{cases} \forall D \in \mathcal{D} & : V(G^*, D) \leq v = V(G^*, D^*) & (A) \\ \forall G \in \mathcal{G} & : V(G, D^*) \geq v = V(G^*, D^*) & (B) \end{cases}$$

for $G^*$ the minimizer of $D_f(p_{\text{data}} || p_G)$ and a $D^* \in \bar{\mathbb{R}}^{|\mathcal{N}|}$. Now applying Lemma 9 we have that

$$\tilde{D} = \text{Best-Response}(G^*) = f'\left( \frac{p_{\text{data}}(x)}{p_{G^*}(x)} \right).$$

Additionally, by the optimality of the response and the consequence (A) of Minimax Theorem it holds that $V(G^*, \tilde{D}) = v$. Finally, assuming that $f$ is strictly convex we get that $V(G, \cdot)$ is strictly concave, Best-Response$(G^*)$ is unique and thus

$$D^* = \tilde{D} = f'\left( \frac{p_{\text{data}}(x)}{p_{G^*}(x)} \right)$$

$\square$

---

[6]This assumption guarantees that the $D_f$ is always finite even if the distribution chosen by the generator is not fully supported on $\mathcal{N}$. This in turn guarantees that $D^*$ is also finite resulting in a meaningful equilibrium. Unbounded divergences like KL are known to be problematic for GANs even in practice Arjovsky et al. (2017).

### 9.1.3 WGAN

The utility of the zero-sum game $V(G, D)$ for the distribution $p_{\text{data}}$ over the discrete metric space $(\mathcal{N}, \text{dist})$

$$V(G, D) = \mathbb{E}_{\mathbf{X} \sim p_{\text{data}}}[D(\mathbf{X})] - \mathbb{E}_{\mathbf{X} \sim p_G}[D(\mathbf{X})]$$
$$= \sum_{x \in \mathcal{N}} (p_{\text{data}}(x) - p_G(x))D(x) \text{ where } \|D\|_{\text{Lip}} \leq 1$$

On the one hand, it is easy to check that for a fixed discriminator $D$, the utility function is linear over the $p_G$ operator. On the other hand, for a fixed generator $G$, the utility function is linear over $D$.

We start our work with the following lemmas

**Lemma 14** (Arjovsky et al. (2017)). *For a fixed generator $G$ the optimal discriminator is a solution of the following linear program*

*maximize over $D(\cdot)$*
$$\sum_{x \in \mathcal{N}} (p_{data}(x) - p_G(x))D(x)$$

*subject to*
$$|D(x) - D(x')| \leq \text{dist}(x, x'), \forall x, x' \in \mathcal{N}$$

*where the optimal value of the LP is the Earth mover's distance between $p_{data}$ and $p_G$.*

*Proof.* Indeed, by definition any solution of the above LP is an optimal discriminator over a fixed generator $G$. To complete the proof of the statement, we recall that Earth Mover's distance of $(p_{\text{data}}, p_G)$ is equal to
$$\min_{\Delta \in \text{Coupling}(p_{\text{data}}, p_G)} \mathbb{E}_{(\mathbf{X}, \mathbf{X}') \sim \Delta}[\text{dist}(X, X')].$$

Now if we consider the dual formulation of the Wasserstein distance, then the Kantorovich duality Evans (1997); Villani (2008) implies that the above linear program consists exactly the dual linear program which computes the Earth Mover's distance. □

**Lemma 15.** *For a fixed discriminator $D$, any distribution supported only on*
$$S_{G_D^*} = \{x \in \mathcal{N} : \forall x' \in \mathcal{N} \quad D(x) \geq D(x')\}$$

*is an optimal generator when it is allowed to choose any distribution over $\mathcal{N}$.*

*Proof.* Observe that for a fixed discriminator, the optimal generator optimizes
$$-\sum_{x \in \mathcal{N}} p_G(x)D(x)$$

since the other term is independent of the generator. Let us define the following
$$D_{\max} = \max_{x \in \mathcal{N}} D(x)$$

Then we have that
$$-\sum_{x \in \mathcal{N}} p_G(x)D(x) \geq -D_{\max}$$

with the equality being true only for distributions supported only on $S_{G_D^*}$. □

**Lemma 16** (Arjovsky et al. (2017)). *The min-max generator is the following distribution*
$$G^* = \arg\min_{G \in \mathcal{G}} \text{EMD}(p_{data}, p_G).$$

*Proof.* We can substitute in $V(G, D)$ the optimal discriminator from Lemma 14. Thus we get
$$V(G, D_G^*) = \text{EMD}(p_{\text{data}}, p_G)$$

By minimizing $V(G, D_G^*)$, the result follows trivially. □

**Lemma 17.** *The max-min discriminator is*

$$\forall x \in \mathcal{N} : D^*(x) = c, \text{ Constant function}$$

*when the generator is allowed choose any distribution over* $\mathcal{N}$,

*Proof.* We can substitute in $V(G, D)$ the optimal generator from Lemma 5

$$V(G_D^*, D) = D_{\max} \sum_{x \in S_{G_D^*}} p_{\text{data}}(x) - D_{\max} \sum_{x \in S_{G_D^*}} p_G(x)$$

$$+ \sum_{x \notin S_{G_D^*}} p_{\text{data}}(x) D(x) - \sum_{x \notin S_{G_D^*}} \underbrace{p_G(x)}_{0} D(x)$$

Observe that for $x \notin S_{G_D^*}$, if $D$ takes more than two values then setting $D$ equal to the highest of the them for all $x \notin S_{G_D^*}$ improves utility. So for an optimal discriminator we would have a single value $D_{\max} > D_{\min}$. In the end we have that

$$x \notin S_{G_D^*} \implies D^*(x) = D_{\min}$$
$$x \in S_{G_D^*} \implies D^*(x) = D_{\max}$$

Observe that for any combination of $D_{\max}$ and $D_{\min}$ with $D_{\max} > D_{\min}$, the constant discriminator $D_{\max}$ has higher utility. Therefore we can focus our attention on the constant discriminator $D_{\text{const}}(x) = D$, where the optimal value is exactly zero.

$$V(G_{D_{\text{const}}}^*, D_{\text{const}}) = 0$$

Finally, it is easy to check that the choice of constant discriminator satisfies trivially the Lipschitz constraints, i.e $|D_{\text{const}}(x) - D_{\text{const}}(x')| = 0 \leq \text{dist}(x, x')$ for any metric function dist. $\qquad\square$

## 9.2 GANs and Hidden Constrained Optimization

In the following section, we will generalize the results of Section 3.2 and Section 3.3 for the case of *vanilla* GAN of Goodfellow et al. (2014a) whose objective is linear-strong-concave where the maximization part is constrained in the distributional simplex. More precisely,

$$\min_{\substack{p_G(x) \geq 0, \\ \sum_{x \in \mathcal{N}} p_G(x) = 1}} \max_{D \in (0,1)^{|\mathcal{N}|}} V(G, D) = \sum_{x \in \mathcal{N}} p_{\text{data}}(x) \log(D(x)) + \sum_{x \in \mathcal{N}} p_G(x) \log(1 - D(x))$$

At a first glance, by rewriting the equivalent Langrangian formulation of the aforementioned constrained min-max problem we can see that strong-concavity property does not hold any more. However our following theorem shows that by exploiting further the structure of Goodfellow et al. (2014a)'s architecture a convergence result is possible.

**Theorem 14.** *Let $V(Gen_{\boldsymbol{\theta}}, Disc_{\boldsymbol{\phi}})$ be Goodfellow GAN as described in Section 4, where $G, D$ use sigmoid activations. Then for a fully mixed distribution $p_{data}$, $(\mathbf{F}(t) = Gen_{\boldsymbol{\theta}(t)}, \mathbf{G}(t) = Disc_{\boldsymbol{\phi}(t)})$ converges to $(p_{data}, \frac{1}{2}\mathbf{1}_{|\mathcal{N}|})$ as $t \to \infty$ under the dynamics of Equation* (1).

*Proof.* Let us write down our original objective

$$\min_{\substack{p_G(x) \geq 0, \\ \sum_{x \in \mathcal{N}} p_G(x) = 1}} \max_{D \in (0,1)^{|\mathcal{N}|}} V(G, D) = \sum_{x \in \mathcal{N}} p_{\text{data}}(x) \log(D(x)) + \sum_{x \in \mathcal{N}} p_G(x) \log(1 - D(x))$$

In order to remove the constraints from the objective above, we plan to make use of a Lagrange multiplier. We remind the reader that since both the discriminator and the generator use the sigmoid activations, we only have to capture the $\sum_{x \in \mathcal{N}} p_G(x) = 1$ constraint. Thus, our equivalent Langragian is:

$$\min_{\boldsymbol{\theta} \in \mathbb{R}^{|\mathcal{N}|}} \max_{\boldsymbol{\phi} \in \mathbb{R}^{|\mathcal{N}|}, \lambda \in \mathbb{R}} L(\mathbf{F}, \mathbf{G}, \lambda) = \mathbf{p}_{\text{data}}^\top \log(\mathbf{G}(\boldsymbol{\phi})) + \mathbf{F}(\boldsymbol{\theta})^\top \log(1 - \mathbf{G}(\boldsymbol{\phi})) + \lambda(\mathbf{F}^\top \mathbf{1}_{|\mathcal{N}|} - 1)$$

where

$$
\begin{aligned}
\mathbf{F}(\boldsymbol{\theta}) &= \begin{bmatrix} f_1(\theta_1) & f_2(\theta_2) & \cdots & f_{|\mathcal{N}|}(\theta_{|\mathcal{N}|}) \end{bmatrix} \\
\mathbf{G}(\boldsymbol{\phi}) &= \begin{bmatrix} g_1(\phi_1) & g_2(\phi_2) & \cdots & g_{|\mathcal{N}|}(\phi_{|\mathcal{N}|}) \end{bmatrix}
\end{aligned}
$$

and $f_i$ and $g_j$ are sigmoid functions and $\theta_i$ and $\phi_j$ are their one dimensional inputs. Let's write again the equivalent dynamics of Equation (3) for the sigmoid activations and the Langrage multiplier. Applying the same steps with Theorem 3 for sigmoids:

$$
\left\{
\begin{aligned}
\dot{f}_i &= - & f_i^2(1-f_i)^2 \frac{\partial L}{\partial f_i}(\mathbf{F},\mathbf{G}) & \quad \forall i \in [|\mathcal{N}|] \\
\dot{g}_j &= & g_j^2(1-g_j)^2 \frac{\partial L}{\partial g_j}(\mathbf{F},\mathbf{G}) & \quad \forall j \in [|\mathcal{N}|] \\
\dot{\lambda} &= & \sum_{i=0}^{|\mathcal{N}|} f_i - 1 &
\end{aligned}
\right\}
$$

Since all initializations are safe in this game, our "generalized" Lyapunov function:

$$
H(\mathbf{F},\mathbf{G},\lambda) = \sum_{i=0}^{|\mathcal{N}|} \int_{p_{\text{data}}(x_i)}^{f_i} \frac{z - p_{\text{data}}(x_i)}{z^2(1-z)^2} \mathrm{d}z + \sum_{j=0}^{|\mathcal{N}|} \int_{1/2}^{g_j} \frac{z - 1/2}{z^2(1-z)^2} \mathrm{d}z + \frac{(\lambda - \lambda^*)^2}{2}
$$

where $\lambda^*$ is the Langrange multiplier at the equilibrium of the non-hidden game and $x_i$ is the $i$-th element of $\mathcal{N}$. Applying the same steps as in Lemma 3 we get that GDA approaches the largest invariant set $E$ of points $(\mathbf{F},\mathbf{G},\lambda)$ that have the following properties

$$
L(\mathbf{p}_{\text{data}},\mathbf{G},\lambda) = L\left(\mathbf{p}_{\text{data}},\frac{1}{2}\mathbf{1}_{|\mathcal{N}|},\lambda^*\right)
$$

$$
L\left(\mathbf{F},\frac{1}{2}\mathbf{1}_{|\mathcal{N}|},\lambda^*\right) = L\left(\mathbf{p}_{\text{data}},\frac{1}{2}\mathbf{1}_{|\mathcal{N}|},\lambda^*\right)
$$

For the first equality, we have that the value of $\lambda$ does not affect $L$ when the generator respects the sum to one constraint. Thus

$$
L(\mathbf{p}_{\text{data}},\mathbf{G},\lambda) = L(\mathbf{p}_{\text{data}},\mathbf{G},\lambda^*)
$$

Then we can observe that $L(\mathbf{p}_{\text{data}},\mathbf{G},\lambda^*)$ is strictly concave in $\mathbf{G}$ and given that $\frac{1}{2}\mathbf{1}_{|\mathcal{N}|}$ is its unique minimum we have that

$$
L(\mathbf{p}_{\text{data}},\mathbf{G},\lambda^*) = L\left(\mathbf{p}_{\text{data}},\frac{1}{2}\mathbf{1}_{|\mathcal{N}|},\lambda^*\right) \implies \mathbf{G} = \frac{1}{2}\mathbf{1}_{|\mathcal{N}|}
$$

Given that $E$ is an invariant set and $\mathbf{G}$ is constant in $E$, we have that $\dot{\mathbf{G}} = 0$. In other words,

$$
0 = \frac{1}{2^2}\left(1 - \frac{1}{2}\right)^2 \frac{\partial L}{\partial g_j}\left(\mathbf{F},\frac{1}{2}\mathbf{1}_{|\mathcal{N}|},\lambda\right) \quad \forall j \in [|\mathcal{N}|]
$$

As a consequence we have that

$$
\frac{\partial L}{\partial g_j}\left(\mathbf{F},\frac{1}{2}\mathbf{1}_{|\mathcal{N}|},\lambda\right) = 0 \implies f_j = p_{\text{data}}(x_j) \quad \forall j \in [|\mathcal{N}|]
$$

Once again, given that $E$ is an invariant set and $\mathbf{F}$ is constant in $E$, we have that $\dot{\mathbf{F}} = 0$

$$
0 = p_{\text{data}}(x_i)^2 \left(1 - p_{\text{data}}(x_i)\right)^2 \frac{\partial L}{\partial f_i}\left(p_{\text{data}},\frac{1}{2}\mathbf{1}_{|\mathcal{N}|},\lambda\right) \quad \forall i \in [|\mathcal{N}|]
$$

This leads to

$$
\frac{\partial L}{\partial f_i}\left(p_{\text{data}},\frac{1}{2}\mathbf{1}_{|\mathcal{N}|},\lambda\right) = 0 \implies \lambda = \log\left(\frac{1}{2}\right) \quad \forall i \in [|\mathcal{N}|]
$$

Observe that by the optimality conditions of the non-hidden game, $\lambda^*$ needs to satisfy the same equation and thus $\lambda = \lambda^*$. Clearly we have that

$$
(\mathbf{F},\mathbf{G},\lambda) \in E \implies (\mathbf{F},\mathbf{G},\lambda) = \left(p_{\text{data}},\frac{1}{2}\mathbf{1}_{|\mathcal{N}|},\lambda^*\right)
$$

Thus the dynamics converge to the unique equilibrium of the hidden game. $\qquad\square$

## 9.3 ZERO-SUM GAMES

We close this section with an application of our regularization machinery in hidden bilinear games. Hidden bilinear zero-sum games were introduced by Vlatakis-Gkaragkounis et al. (2019) and they are formally defined as:

**Definition 11** (Hidden Bilinear Zero-Sum Game). *In a hidden bilinear zero-sum game there are two players, each one equipped with a smooth function $\boldsymbol{F} : \mathbb{R}^n \to \mathbb{R}^N$ and $\boldsymbol{G} : \mathbb{R}^m \to \mathbb{R}^M$ and a payoff matrix $U_{N \times M}$ such that each player inputs its own decision vector $\boldsymbol{\theta} \in \mathbb{R}^n$ and $\boldsymbol{\phi} \in \mathbb{R}^m$ and is trying to maximize or minimize $r(\boldsymbol{\theta}, \boldsymbol{\phi}) = \boldsymbol{F}(\boldsymbol{\theta})^\top U \boldsymbol{G}(\boldsymbol{\phi})$ respectively.*

For the special case of hidden bilinear games, Vlatakis-Gkaragkounis et al. (2019) proved that if the dimension of the game is greater or equal than two like (e.g. akin to Rock-Paper-Scissors) then GDA dynamics tend to "cycle" through their parameter space with an even more complex behavior than a typical periodic trajectory. Specifically, the system is formally analogous to Poincaré recurrent systems (e.g. many body problem in physics). In contrast, leveraging Theorem 6, we know that by adding a small regularization term we can "break" the cycling behavior and converge to an approximate Nash Equilibrium. We close this section by presenting a comparison between the optimization portraits of GDA dynamics with the absence or not of a regularization for the archetypical game of Rock-Paper-Scissors:

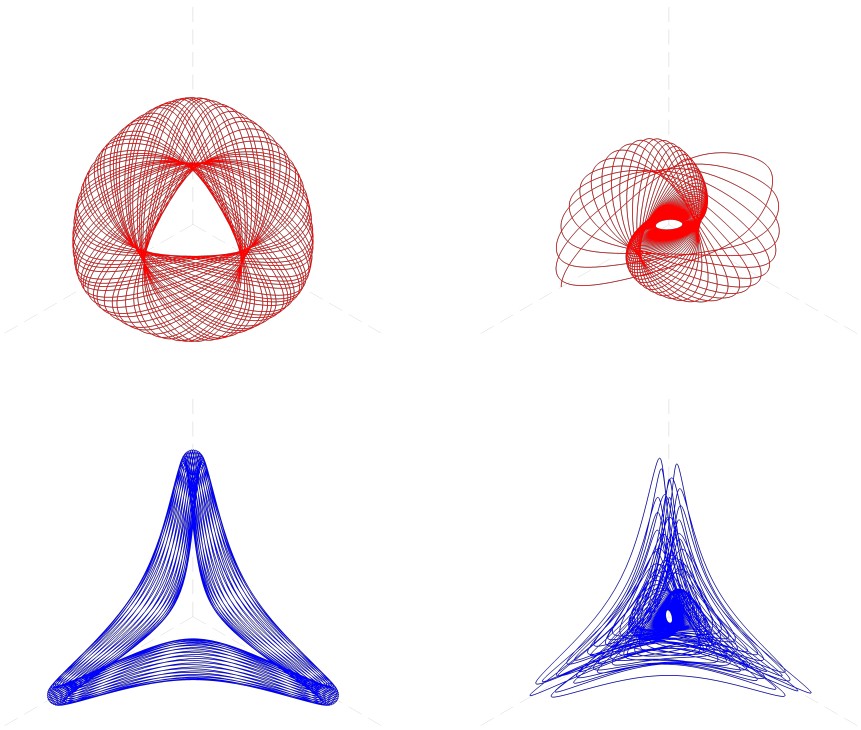

Figure 5: Trajectories of a single player using gradient-descent-ascent dynamics for a hidden bilinear game $L(\mathbf{F}(\boldsymbol{\theta}), \mathbf{G}(\boldsymbol{\phi})) = \mathbf{F}^\top(\boldsymbol{\theta}) A \mathbf{G}(\boldsymbol{\phi})$ where $A$ is the classical Rock-Paper-Scissors table and $\mathbf{F}, \mathbf{G}$ have the sigmoid activations. The two left figures present the Poincaré recurrence for different initializations of the dynamics, a behavior consistent with the Lyapunov stability of Theorem 2. On the other hand, the two figures on the right illustrate convergent to the mixed Nash equilibrium executions which exploit the regularization tools as described in Section 3.3. The regularization terms added are centered at the mixed equilbrium of the game, leading to convergence to the unmodified equilibrium of the Rock-Paper-Scissors game.

