# OpenReview forum: "Solving Min-Max Optimization with Hidden Structure via Gradient Descent Ascent"
_ICLR.cc/2021/Conference — Reject_

### Official Review · AnonReviewer1 · 2020-10-20

**Rating:** 4
**Confidence:** 3

**Review:**

In this paper, the authors introduce a class of games called Hidden Convex-Concave where a (stricly) convex-concave potential is composed with smooth maps. On this class of problems, they show that the continuous gradient dynamics converge to (a neighbordhood of) the minimax solutions of the problem. This is an exploratory theoretical paper which aims at better capturing the behaviors that can be observed e.g. in the training of GANs.


While the paper has merits, it fails in my opinion to clearly explain its theoretical grounds and findings to a non-specialist of continuous dynamics (like me). I thus cannot recommend acceptance for this version based on my comprehension.


* The authors consider the gradient descent/ascent as a standard optimization dynamics. However, in min-max optimization (and even in practical GAN training see e.g. "Reducing Noise in GAN Training with Variance Reduced Extragradient" by Tatjana Chavdarova, Gauthier Gidel, François Fleuret, Simon Lacoste-Julien), extragradient techniques are actually more often used that gradient descent-ascent due to they better theoretical properties (see the first paragraph of the introduction). Notably, EG can converge to a saddle point in bilinear games where gradient fails (see Chavdarova et al. above or Hsieh et al. 2020) even though they globally come from the same continuous-time dynamics. Thus, it is natural to wonder why/if the continuous-time gradient dynamics bears valuable information about the saddle-point problem and associated behaviors. I would like to see the authors address this in the introduction.

* The paragraph "Our class of games" and "Our solution concept" quite verbose and hard to understand (maybe adding in equations the definitions of Nash Eq. and Von Neumann solutions would help). Since a goal of the paper is to provide intuition on the dynamics of certain games with respect to certain solution, these should be made very clear for the reader, this is not the case at the moment.

* I am concerned with the definition of "safe initialization" (Def 3) and notably how it can be checked (which is unfortunately not discussed). Typically, most results assume a safe initialization which looks like a local basin of attraction. This kind of contradicts the statement that the results are "non-local" in the abstract.

* Concerning the difference between the considered setup and previous ones: i) Apart from the "reparametrization" of Section 2.3, what changes between this setup and the convex-concave saddle point gradient dynamics?  ii) and Compared to hidden bilinear games?

* The authors claim in the conclusion that the "modular structure" of their proofs make HCC games suitable "theoretical testbeds". However, it is a bit hard while reading the paper to pinpoint the mathematical tools that can actually be used for further studies. While looking at the 23 pages appendix, I feel that Eq. (1) -- the dynamics; Theorem 1 -- the reparametrization; and Lemma 2 -- derivation of the non-increasing potential, are the most important results (or rather their instantiation for GDA dynamics). In th present version, they are a bit lost in the very long appendix. Maybe dropping down the general case and the regularization parts would enable to better present these points in the paper.


Minor Comments:
* In the first paragraph of the introduction, the authors cite 16 references at once. I am not sure that this is actually informative. Either the authors could break down these refs into smaller chunks/units that relate to one particular kind of results (as id done in the second paragraph) or drastically reduce the number of papers cited there.
* The term "hidden convexity" may be a bit confusing since it may refer to e.g. local strong convexity around the solution. In the literature, the term convex composite problem is sometimes used to denote minimization problems of the form \min_x h(c(x)) with h a convex function and c a smooth mappings (see e.g. "Efficiency of minimizing compositions of convex functions and smooth maps" by Drusvyatskiy and Paquette), this may be mentioned as well.
* In Lemma 2 and Theorem 2, "is a safe for" probably misses the word "initialization".
* In Theorem 2, the sense of "stable" here should be defined.
* Lemma 3: "is the non-empy" should be "be the non-empty".
* bottom of p7: "regularization" is spelled wrong, "it can posed" -> "be posed"
* In Figure 3, how is the continuous dynamics discretized?

---

> ### Author Response · Authors · 2020-11-16
> **Response to AnonReviewer1 (2/2)**
>
>
> The reviewer also has some questions about how one can check if an initialization of an HCC game is safe based on $\textbf{Definition 3}$, which is a sufficient condition for convergence to a global Nash equilibrium of the game. Even in non-convex optimization, a significantly simpler problem compared to non-convex non-concave min-max optimization, understanding which initializations lead to a global minimum of the loss function under gradient descent is a very hard problem. Despite the special structure of the HCC game, understanding which initial conditions are safe still remains at least just as hard. For example, if $f_i$ is a deep neural net and the $p_i$ of the target von Neumann solution happens to be a value close to the global minimum of $f_i$, then we cannot efficiently decide a-priori (e.g. without running GDA) if an initialization is safe.
>
> Having some form of assumptions on the initialization of the game is completely necessary to avoid negative blanket conclusions for most established classes of non-convex non-concave games. Instead of reiterating negative results based on unfortunate initializations that are not in agreement with the empirical success of GANs, we choose to study the dynamics of HCC games under safety.
>
> It is important to highlight though that regions of attractions that are possible under safety are not similar to standard results of local convergence and this is why we describe the convergence type as “non-local”.  Indeed, in a local convergence result the algorithm needs to be initialized in a small basin of the equilibrium to converge to it. This limitation does not apply to our results where depending on the choice of $F$ and $G$, the points that converge to an equilibrium can be arbitrarily far away from it. For example, observe that for the function of $\textbf{Figure 1}$, at least with respect to $f_i$, all negative initializations except the local minima and maxima of $f_i$ are safe. Clearly, this is a non-local convergence result.
>
> The reviewer is also unsure about the differences of our work with setups studied in prior work. The class of HCC games is a generalization of both convex-concave games and hidden bi-linear games. Convex-concave games correspond to $F$ and $G$ being the identity operators whereas hidden bi-linear games correspond to $L$ being a bi-linear map. We refer the reviewer to our response to $\textit{AnonReviewer4}$ for a comprehensive analysis of the technical challenges of HCC as compared to the work of [[VGFP19](https://arxiv.org/pdf/1910.13010.pdf)] on hidden bi-linear games as well as compared to the convex-concave games.
>
> We understand the reviewer’s concern that a significant portion of technical details remains in the Appendix. The combination of the dynamics of $\textbf{Eq. (1)}$, the reparametrization theorem as well as the potential function are necessary tools to even establish the expected behavior of $\textbf{Eq. (3)}$. But transferring the results to the dynamics of $\textbf{Eq. (1)}$ still requires some additional tools that are found in the corresponding proofs in the Appendix.
>
> To guide the reader we have tried to provide summaries of what is being proven in each theorem in the Appendix as well as the basic idea of the arguments used. Unfortunately moving this discussion to the main paper is not straightforward as one can observe based on the extensive background material contained in $\textbf{Section 7}$. In response to the reviewer’s comment, we will do our best to include additional intuition about the technical tools required for each proof

---

> > ### Comment · AnonReviewer1 · 2020-11-20
> > **Some quick comments about the response**
> >
> > Thank you for the response and the clarifications. I was indeed confused about some points such as the discretization of the continuous time dynamics and the relation between Von Neumann solutions and Nash Equilibria. However, I'm afraid the same may happen to non-specialists reading the paper so I would appreciate a bit more pedagogy in the first parts of the paper.
> >
> > I agree that we have to lay out some assumptions to work, I was mainly wondering if this "safe initialization" could be related to other "assumptions" of the literature in some special cases (either from the study of GDA in continuous or discrete-time). Typically, the paragraph "It is important to highlight" in your response kind of clarifies it (and could be added in some form in the paper), but I still fail to see if this connects with other assumptions of the literature.
> >
> > Could you also comment on discretization ?

---

> > > ### Author Response · Authors · 2020-11-21
> > > **Other assumptions & Safety Condition**
> > >
> > > We would like to thank the reviewer again for his willingness to give us feedback to improve the readability of our work to non-continuous time experts. We commit to include the crucial part of our discussion about the various clarifications in the first sections of the camera-ready version.
> > >
> > > AnonReviewer 1 asked about a comparison between our safety assumption and the ‘‘other” assumptions in the literature. Roughly speaking, our assumptions are totally orthogonal to the typical assumption about convergence in min-max settings. Critically, our results are non-local, whereas prior results in the literature are local and hence the related assumptions reflect sufficient conditions for applying variants of the Stable Manifold Theorem (SMT) which themselves relate to system eigenvalues.
> > >
> > > On the other hand, our safety conditions address mainly the issue of the implementability of the equilibrium of $L(x,y)$ when it is composed with the non-convex operators $F(x), G(y)$. Clearly these two types of assumptions lie in orthogonal directions (the curvature around the equilibrium VS the range of the operators).
> > >
> > > Finally, the reviewer asked about extensions of our result to non--continuous methods and possible discretizations. There are multiple promising approaches to finding Nash Equilibria or Von-Neumann solutions with discrete dynamics. One methodology comes from the seminal work of Benaim 1990 ([Dynamics of Stochastic Approximation Algorithms](http://www.numdam.org/article/SPS_1999__33__1_0.pdf)) and another recent one is of  [Haihao Lu](https://arxiv.org/abs/2001.08826) about the $O(s^r)$-Resolution ODE Framework for Discrete-Time Optimization Algorithms and Applications to the Linear Convergence of Minimax Problems, which can give us a promising path for a discrete-time analysis. Leveraging those frameworks introduces extra technical hurdles and concepts (e.g. pseudotrajectories or resolution ODEs), which lie beyond the scope of this work. Nevertheless, we believe that our continuous-time results will serve as an important building block of any argument in that direction.
> > >
> > > We will be happy to incorporate this discussion in our manuscript and we look forward to any other suggestions that you may have.

---

> ### Author Response · Authors · 2020-11-16
> **Response to AnonReviewer1 (1/2)**
>
>
> We would like to thank the reviewer for the insightful comments on how to make our work more approachable and understandable by a wider audience.
>
> The reviewer feels that we implied that alternative optimization algorithms like extragradient dynamics are non-standard or secondary. This was not our intention when we mentioned that GDA is a standard optimization algorithm for min-max optimization. We were merely referring to the fact that GDA is a foundational algorithm from which other gradient-based dynamics can be derived. Thus studying GDA for HCC is an important stepping stone towards non-local arguments about other dynamics in non-convex non-concave games.
>
> We also feel that the reviewer has made some inaccurate statements about the connections of discrete-time EG and GDA to the continuous-time GDA. It is not the case that standard and “optimistic”/extra-gradient techniques correspond to the same continuous-time dynamical system.
> For example, [Cheung et al.](https://papers.nips.cc/paper/2020/file/66de6afdfb5fb3c21d0e3b5c3226bf00-Paper.pdf) recently studied both Multiplicative Weights Update (MWU) and Optimistic (MWU), which are known to be divergent and convergent respectively in bilinear zero-sum games via their connections to their continuous-time analogues and showed that the positive convergent behavior OMWU can be understood by the properties of its continuous-time method, which is a new ODE and not the well-known continuous-form of MWU (replicator dynamics). Thus, it should be clear that continuous-time dynamics bear very valuable information about the behavior of algorithms in saddle point problems.
>
> The reviewer is also confused about the relation of von Neumann solutions and Nash equilibria in the context of HCC games. For the convex-concave $L$, von Neumann solutions correspond to inputs $p$ and $q$ such that $L(p,q) = \min\max L(x, y)$. Notice that since this corresponds to a zero-sum competition, they coincide with the Nash equilibria of $L$. Our solution concept for HCC games are $(\theta^*, \phi^*)$ such that $F(\theta^*)=p$ and $G(\phi^*)=q$ for some von Neumann solution $(p,q)$ of $L$. The chosen solution concept is not arbitrary since clearly
> $(\theta^*, \phi^*)$ is a global Nash equilibrium of the HCC. We will try to make this more clear in our discussion.

---

### Official Review · AnonReviewer2 · 2020-10-26
**A paper with potential impact despite several important limitations**

**Rating:** 6
**Confidence:** 4

**Review:**

### Summary

With the aim of improving our understanding of GAN training, the paper studies the behavior of gradient descent ascent (GDA) dynamics in a subclass of the so-called _hidden convex-concave games_. In this family of problems, the players control the input variables of smooth functions whose outputs are taken as inputs of a convex-concave game.
For the problems covered by this paper, different types of stability around nash equilibria of the hidden game are proved under different assumptions. In particular, the concept of _safe_ initializations is introduced, which clearly distinguishes between the case where the trajectory can and cannot converge to a hidden game solution. As an example, the authors prove that when the hidden game is strictly convex-concave, with safe initialization GDA dynamics converge to a point that solves the hidden minimax game.

### Pros

This paper can have an impact on the analysis of GAN optimization by examining a different line of attack on the problem.
The analysis and the presentation are clean and the appendix is extremely well-organized.
The importance of the initialization is further highlighted by the concept of safety, reflecting the additional complexity caused by the nonlinear transformation.
Overall, this paper presents some beautiful results for the problem under study, and provide concise explanations on how these results are obtained.

### Cons / Limitations of the paper

There are however several important limitations of the work which make it difficult to evaluate the significance of the results.

#### Slight overclaim on the problems covered by the paper
Up to page 2, one may believe that the paper addresses the general hidden convex-concave game as defined by (HCC). Then, on page 3, it turns out actually the paper only focuses on a special type of HCC for which every coordinate of the function output is independently determined by its input variables. This seems to be rather restricted and it is unclear how the results of the paper can be applied to general HCC. Moreover, this limitation is mentioned nowhere in the abstract or the introduction of the paper.

#### Ambiguities in how GANs fit the framework
While the authors claim that GAN training is a specific case of HCC, it is not straightforward from the text that this in fact indicates either of the following
1. The probability distribution has finite discrete support.
2. The discriminator outputs a function and the generator outputs a measure.

The paper (and Section 9 in particular) centers on case 1 which is quite far from real GAN problems, and this message is not so clear in the paper (with only the keyword "discrete" mentioned discreetly). On the other hand, for the second case, we need to study minimax games in a Banach space which is not covered by the paper.

#### The potential of the approach is unclear

Finally, in addition to the above two limitations which weaken the link between the problems studied by the paper and real GAN training, the discretization from continuous ODE to practical algorithms is also non-trivial. The study of the dynamics itself is surely of interest. Nonetheless, taking all these into account, I am just wondering if it would really be possible/suitable to study GAN optimization through the lens of hidden convex-concave games.

### Score justification

I appreciate the efforts that the authors have made to come out with all the results and to present them in such a succinct manner. I however do not put a higher score due to the concerns that I raised above.

---

> ### Author Response · Authors · 2020-11-16
> **Response to AnonReviewer2 (1/1)**
>
> We thank the reviewer for his valuable comments and thorough review.
>
> Regarding the comment of $\textit{AnonReviewer2}$ about a potential overclaim in our work, we would like to highlight that the restriction on HCC where $f_i$ and $g_j$ have all disjoint inputs is already clearly mentioned at the second line of the $\textbf{Our results}$ paragraph (page 2). Since other reviewers did not make a similar comment, it is possible that $\textit{AnonReviewer2}$ might have missed this.
>
> $\textit{AnonReviewer2}$ expressed some confusion about how training GANs fit the HCC framework. We would like first to remind the reviewer that although we argued in the Applications section that some instances of training GANs fit the HCC framework, we did not claim that all GAN training can be viewed as an HCC game. Despite this, we believe that the tools developed here to transfer results from the operator space of $F$ and $G$ to the parameter space, which are the first of their kind to the best of our knowledge, are a major stepping stone.
>
> That being said, we would like to point out that GANs can also be applied in settings that are not captured by either cases brought forth by the reviewer. Specifically, for some settings, the generator does not output a measure directly but outputs the parameters of a distribution. The application of [Lei et al](https://arxiv.org/pdf/1910.07030.pdf). on WGANs is an instance of such an application. The generator does not learn from individual samples of the distribution directly but instead tries to match the moments of the target distribution. The setting is more general than traditional moment matching since the observed samples may be transformed by non-linear activations. HCC games capture this class of GAN applications directly.
>
> Finally, $\textit{AnonReviewer2}$ also inquired about promising approaches to finding Nash Equilibria or Von-Neumann solutions with discrete dynamics. Focusing on safe initializations, $\textbf{Theorem 5}$ & $\textbf{Corollary 1}$ in combination with the seminal work of Benaim 1990 ([Dynamics of Stochastic Approximation Algorithms]( http://www.numdam.org/article/SPS_1999__33__1_0.pdf )) can give us a promising path for a discrete-time analysis. Leveraging Benaim’s framework introduces extra technical hurdles and concepts (e.g. pseudotrajectories), which lie beyond the scope of this work. Nevertheless, we believe that our continuous-time results will serve as a fundamental building block of any argument in that direction.

---

> > ### Comment · AnonReviewer2 · 2020-11-19
> > **Some remarks**
> >
> > Thank you for the reply.
> >
> > The first two points that I mentioned were equally brought up by Reviewer 3 and my main concern is indeed on how things are presented in its current form (I acknowledge that I probably overlooked the word "disjoint" on page 2 so it may worth emphasizing it a bit more; in the following I will rather focus on the GAN part).
> >
> > In my opinion, even the generator is outputting the parameter of a distribution it is still learning a distribution, but this problem is not relevant here. What is more important is that the problems highlighted in the Applications sections (main and appendix) concern only toy GAN models (Gaussian distribution + linear discriminator or probability on a finite set) and this is not so clear from the text.
> > That is why I am suggesting distinguishing clearly between the two cases: the general GAN with a HCC in infinite-dimensional space and some toy GAN models such as the examples mentioned in the paper that actually fit the analysis of the work. Stating this way, it would be clear that the high-level concept indeed applies to very general problems (as briefly discussed by the authors in the last paragraph of section 4, see also https://arxiv.org/abs/2002.05820), and this work is addressing a simplified version of the problem which encompasses several toy GAN models.
> >
> > To finish, I agree with Reviewer 1 that different discretization of a same continuous dynamic may lead to different behaviors. Indeed, optimistic methods can be understood as deriving from a different continuous dynamic as also highlighted in https://arxiv.org/abs/1905.10899. Nonetheless, as shown in https://arxiv.org/pdf/2006.09065.pdf, these very different methods can also come from discretization of a single continuous dynamic system. In particular, I am curious to know if the authors can provide any reference for a different continuous dynamic of extragradient.

---

> > > ### Author Response · Authors · 2020-11-21
> > > **Infinite-dimensional & Extra-Gradient**
> > >
> > > We thank the reviewer for his valuable comments and thorough review and additional references that we will happily incorporate.
> > >
> > > We certainly agree that our result does not apply in the infinite-dimensional case. We definitely did not mean to suggest that this is the case and we believe that a reader even after a cursory pass over the paper should have little trouble telling that this is not so (ie Banach spaces are never mentioned). On the other hand, as we pointed out in our very recent response to **AnonReviewer 3** our framework covers applications of WGANs that go beyond the scope outlined above, namely parameter matching over transformed distributions.
> > >
> > > In any case, we agree that it is a good idea to explicitly point out the case of the infinitely dimensional HCC games as the ultimate frontier of this type of work. Naturally, this goal is well beyond scope of this work. Our contribution is a modest step towards this very ambitious direction, which sets up some early foundational work including the first non-local asymptotic stability result for any class of HCC games.
> > > As we pointed out in our initial response to **AnonReviewer 3** these settings lie truly beyond pseudo-monotone games and to our knowledge represent the most advanced version of global convergence result in min-max games. We will be happy if the reviewer could point out any other global convergence result in a class of nonconvex non-concave games that was a closer representative of GANs than our setting. For example, https://arxiv.org/abs/2002.05820 despite its somewhat similar themes that the reviewer nicely points out offers no such analysis.
> > >
> > > **AnonReviewer 2** also inquired about the existence of different ODE for the extra gradient. A recent work of  [Haihao Lu](https://arxiv.org/abs/2001.08826) describes a $O(s^r)$-Resolution ODE framework, where different discrete-time algorithms (DTA) correspond to different continuous dynamics. Thus by choosing a small enough resolution parameter, each DTA shares different local stability-convergence conditions. For the case of extra gradient check Corollary 1 (item iii) and v)) and Proposition 1.
> > >
> > > We hope that these address the questions of the reviewer satisfactorily.

---

### Official Review · AnonReviewer4 · 2020-10-28
**This work is a strictly convex-concave extension of (Vlatakis-Gkaragkounis et al. 2019).**

**Rating:** 5
**Confidence:** 4

**Review:**

This paper studies min-max problems of the form L(F(x),G(y)), where L is a strictly convex-concave loss. The authors prove that, under certain assumptions, the GDA dynamics converges to von Neumann solutions. Toy examples are provided to illustrate the usage of the framework.

I see this paper as a rather incremental extension of the work (Vlatakis-Gkaragkounis et al. 2019). Specifically, Lemma 1 appeared in (Lemma 1, Vlatakis-Gkaragkounis et al. 2019), Theorem 1 appeared in (Lemma 2, Vlatakis-Gkaragkounis et al. 2019), Definition 3 makes a strong assumption that basically gets rid of the challenges of non-convexity, and Lemma 2 appeared in (Theorem 2, Vlatakis-Gkaragkounis et al. 2019).

The key difference between this work and (Vlatakis-Gkaragkounis et al. 2019) is the strict convex-concave losses (present work) vs. bilinear losses (Vlatakis-Gkaragkounis et al. 2019). Due to this difference, the function (4) in Lemma 2 is time-invariant (or acts as a Hamiltonian) in (Vlatakis-Gkaragkounis et al. 2019), whereas in this submission it is non-increasing in the presence of strict convexity, and hence enabling Lyapunov-type analysis. This is an almost immediate conclusion one can draw from studying (Vlatakis-Gkaragkounis et al. 2019), and hence the contribution does not seem to meet the standard of top ML venues.

In addition, the authors studied unconstrained min-max problems with hidden structures, while in Section 4 they claimed that GANs can be viewed as **constrained** min-max problems. I would like to point out that the presence of constraints is a significant difference as it would completely invalidate all the analysis in this paper. The toy example of WGAN is also not convincing enough as it is tied to one dimensionality with a (practically irrelevant) quadratic regularizer.

In conclusion, I did not see much novelty of this submission in terms of theory, and there is no practical implication. I hence suggest rejection for the Area Chair.

---

> ### Author Response · Authors · 2020-11-16
> **Response to AnonReviewer4 (2/2)**
>
> About the constrained optimization, we would like to clarify that we never claimed that our theorems, as stated, cover the general case of constrained min-max optimization. On the other hand, we were very fastidious by providing proofs tuned for the experiment of $\textbf{Figure 3}$ which includes a sum-to-1 constraint. You can find the proof in $\textbf{Theorem 14}$. More broadly, an HCC game with smooth convex equality constraints in the output space can be turned to an unconstrained HCC game through the use of Lagrange multipliers.
>
> Regarding our WGAN experiment, we feel that the reviewer has misinterpreted the intent of the discussion. We are highlighting how our results on HCC games can readily provide direct insights on the results of prior work on GANs.
> [Lei et al.](https://arxiv.org/pdf/1910.07030.pdf) proposed to add a quadratic regularizer to the WGAN objective, arguing that this was merely a replacement for the Lipschitz constraints of the standard GAN formulations. Even for this toy 1D setting, that is chosen for visualization purposes, our results in conjunction with the ones of [[VGFP19](https://arxiv.org/pdf/1910.13010.pdf)], directly predict that the quadratic regularizer is far from irrelevant and is actually necessary to ensure convergence. Observe that our results directly apply to any differentiable activation function. The quadratic activation was chosen because we can compute the expectations and $H$ analytically to produce the left plot of $\textbf{Figure 4}$.

---

> ### Author Response · Authors · 2020-11-16
> **Response to AnonReviewer4 (1/2)**
>
> $\textit{AnonReviewer4}$ has concerns about the novelty of our work. Indeed, the main predecessor of [our work](https://openreview.net/pdf?id=e3KNSdWFOfT) is [[VGFP19](https://arxiv.org/pdf/1910.13010.pdf)], which first introduced the framework of hidden games. While we use tools regarding reparametrization and safety from their work, the rest of our analysis and the technical ideas behind them are qualitatively different.
>
>
> As we highlighted in the introduction, the challenges of HCC games are twofold. First, the game dynamics in the operator space of $F$ and $G$ do not correspond to a simple convex concave game. Additionally, it is still not clear how to transfer stability/convergence properties from the operator space of $F$ and $G$ to the parameter space of $\theta,\phi$. More concretely, it is unclear how to transfer results from the dynamical system of $\textbf{Eq. (3)}$ to the one of $\textbf{Eq. (1)}$.
>
> Any similarities with [[VGFP19](https://arxiv.org/pdf/1910.13010.pdf)], that the reviewer highlighted have to do with the first challenge. In contrast, in terms of the second challenge, there is a substantial gap between our work and the results of [[VGFP19](https://arxiv.org/pdf/1910.13010.pdf)]. In [[VGFP19](https://arxiv.org/pdf/1910.13010.pdf)], the main claim of Poincaré recurrence in hidden bi-linear games can be transferred to the dynamical system of $\textbf{Eq. (1)}$ only when each $f_i$ and $g_j$ are globally invertible. To understand how limiting this requirement is, all $n_i$ and $m_j$ need to be one for this to hold. In contrast, observe that all of our results, except $\textbf{Theorem 3}$ which is shown purely for exposition purposes, do not have this requirement.
>
> To the best of our knowledge, there has not been any related work studying the transfer of stability results between dynamical systems in different spaces that do not rely on the existence of one to one maps between the spaces.
>
> Let us show through a concrete example of why the transfer of stability statements given a Lyapunov function in the output space is non-trivial. $\textbf{Lemma 2}$ provides an example of a non-increasing function in the operator space that is not sufficient to prove any stability property for convex-concave games in the parameter space. Indeed, observe that $\theta_i$ can move on a level set of $f_i$ arbitrarily away from an equilibrium without ever-increasing the value of $H$.
>
>
> Although $\textbf{Theorem 4}$ does not prove stability in the parameter space, it still provides guarantees about the stability of the outputs $F$ and $G$. Once again, observe that $\textbf{Theorem 4}$ and $\textbf{Theorem 2}$ provide very different guarantees since $\textbf{Theorem 4}$ talks about sets of initializations of $\textbf{Eq. (1)}$ that map to a potentially infinite amount of different systems of the form of $\textbf{Eq. (3)}$. $\textbf{Theorem 4}$ provides novel sufficient conditions for constructing neighborhoods of safe initializations that did not appear in [[VGFP19](https://arxiv.org/pdf/1910.13010.pdf)]. It also leverages a potentially infinite amount of Lyapunov functions of many coupled dynamical systems in the $F$ and $G$ space to provide its hidden stability guarantees. To the best of our knowledge, this result is the first of its kind.
>
> Similar challenges need to be addressed when going from $\textbf{Lemma 3}$ to $\textbf{Theorem 5}$. Each initialization of $\textbf{Eq. (1)}$ corresponds to a single initialization of the dynamical system of $\textbf{Eq. (3)}$. It is clear once again that  $\textbf{Lemma 3}$ is not by itself sufficient to guarantee convergence of any initial condition of $\textbf{Eq. (1)}$. Our novel analysis of the region of attraction based on the study of the sublevel sets of $H$ is thus an integral part of the proof of $\textbf{Theorem 5}$. Analyzing the boundedness of sublevel sets of $H$ is challenging since $X_{\theta_i(0)}$ and $X_{\phi_j(0)}$ are based on existential results.
>
> Finally, for $\textbf{Theorem 7}$, it is clear that the standard strong Lyapunov function argument of strongly convex-concave games does not directly apply here. To sidestep this we provide novel initialization dependent upper and lower bounds of $H$ that did not appear in [[VGFP19](https://arxiv.org/pdf/1910.13010.pdf)],.
>
> To the best of our knowledge, our work is the first one to provide sufficient conditions for non-local convergence to a game theoretically meaningful solution for a wide family of non-convex non-concave min-max problems exploiting these tools in such a way. We will happily incorporate part of this discussion to illustrate the differences between our work and the initial work of [[VGFP19](https://arxiv.org/pdf/1910.13010.pdf)],.

---

> > ### Comment · AnonReviewer4 · 2020-11-20
> > **Some remarks**
> >
> > I thank the authors for the thorough reply.
> >
> > -  "In [VGFP19], the main claim of Poincaré recurrence in hidden bi-linear games can be transferred to the dynamical system of  only when (1) each $f_i$ and $g_i$ are globally invertible.... (our results) do not have this requirement."
> >
> > This is again something that can be readily read off from [VGFP19]: to study recurrence you naturally need global invertibility, whereas in the strictly convex-concave case you only want the solutions in the output space to be "covered" by the hidden functions (F and G). For instance, you can have $F(\theta)=F(\theta') = $ solution, thus invertibility is not a concern.
> >
> > I understand this is a difference between this work and [VGFP19], and I also understand the contributions of Theorem 4, 5, 7. My contention is that this work does not bring much new beyond [VGFP19], since the "safe initialization" and the H function ideas are already present in [VGFP19]. Generalizing them from bilinear to strictly convex-concave seems straightforward and does not bring much novelty.
> >
> >
> > - On the other hand, we were very fastidious by providing proofs tuned for the experiment of Figure 3...
> >
> > Indeed, I missed Theorem 14; my apology.
> >
> > Theorem 14 applies only to sigmoid activation which is quite restrictive and uncommon in practice. Hence, the practical value is still limited.
> >
> > - Regarding our WGAN experiment, we feel that the reviewer has misinterpreted the intent of the discussion.
> >
> > I think my original comment was misleading: I meant to say that I cannot see the practical value of this work based on the provided experiments, including the WGAN example (which is more of a theoretical nature).

---

> > > ### Author Response · Authors · 2020-11-21
> > > **Response about the novelty concerns**
> > >
> > > It is unclear what the reviewer means when he says that the requirement of invertibility can be read off from  [[VGFP19](https://arxiv.org/pdf/1910.13010.pdf)]. The reason why  [[VGFP19](https://arxiv.org/pdf/1910.13010.pdf)] cannot transfer the recurrence results from the output space to the input space is because the Poincare recurrence theorem guarantees recurrence in the output space for almost all initializations but not all of them. If the Poincare recurrence theorem had a guarantee for all initializations then the transfer would be immediate. The distinction between almost all and all initializations for the Poincare recurrence theorem is a very complex subject that is not yet formally understood in the literature. We thus believe that conclusions about when invertibility is required or not cannot just be “read off” from a paper or be resolved by intuitive arguments directly.
> > >
> > > The situation above directly mirrors our discussion about contribution in $\textbf{Theorem 5}$, where $\textbf{Lemma 3}$ only provides convergence guarantees for some initializations of the $\textbf{Eq. (3)}$ dynamical systems and not all of them. We avoid going for black-box style reductions through global invertibility by providing a novel analysis for the structure of the sublevel sets of $H$. Our work broadens the toolbox of parameter space - output space transfer results significantly.
> > >
> > > Regarding $\textbf{Theorem 14}$, the sigmoid activations are used as a concrete example to match the experiment we did in $\textbf{Figure 3}$. Essentially, what is required by $\textbf{Theorem 14}$ is that the $f_i$ and $g_j$ of the generator and discriminator to map in (0,1) and the initialization of the dynamical system to be safe for the equilibrium of the hidden game. Sigmoid activations are chosen as examples because they satisfy the restriction for all potential initializations and target distributions.
> > >
> > > To sum up, our aim in this work is to make the first steps towards constructing a theoretically tractable framework that captures many of the complexities of GAN training. We firmly believe that leveraging the special structure of GAN training objectives is a necessary step to avoid reaching blanket negative conclusions that do not agree with the empirical success of GANs.

---

### Official Review · AnonReviewer3 · 2020-10-28
**Interesting progress beyond convex concave games. Application to GANs questionable?**

**Rating:** 5
**Confidence:** 4

**Review:**

Summary:
While single-agent optimization is quite well understood and even convergence results in the nonconvex setting, the study of non-convex-concave saddle point problems is still in its infancy. In particular, recent work by Letcher (2020) and Hsieh et al (2020) suggests that even many recently proposed modifications of simultaneous gradient descent are not guaranteed to converge in the non-convex-concave case.
The present work makes significant progress on this problem by introducing hidden convex-concavity, a class of structured problems nonlinear functions $F(\theta)$ and $G(\phi)$ that depend on only one of the agents $\theta$ and $\phi$, are plugged into a convex-concave problem $L(\cdot, \cdot)$, resulting in $\min_{\theta} \max_{\phi} L(F(\theta), G(\phi))$.
Here, the components $F_i$ and $G_j$ of $F$ and $G$ are multivariate, real-valued functions $f_i$, $g_i$ of disjoint sets of components of $\theta$ and $\phi$.
By cleverly relating the dynamics of the $f_i$,$g_i$ to those of the $\theta_j$, $\phi_j$, the authors derive a Lyapunov function for the underlying dynamics and use it to prove asymptotic stability and convergence under very minor additional assumptions.
They furthermore show a notion of "hidden convergence" of the $f_i$, $g_i$ that is particularly relevant to the overparametrized regime.
In the last section, the authors relate their findings to GANs.

Decision:
I believe that this is good work that will be interesting to a wide range of readers. My only concern is the following:

In the definition of hidden convex-concavity, the functions $f_i, g_j$ are multivariate and real valued, but depend on **disjoint** sets of parameters. In particular, it does not seem to cover the case of  $L(f(\phi), g(\theta))$ where $f$ and $g$ are multivariate vector valued functions.
However, this is arguably the case in GANs, where the discriminator and generator can be thought of high/infinite-dimensional "vectors" that depend nonlinearly on all parameters.
Therefore, the results of the authors do not seem to apply to GANs even formally? Please let me know if there is something that I overlooked.

I believe that the results are interesting and relevant even if they would not apply to GANs. However, in this case the application section and motivation of the paper might need some restructuring.

Other suggestions/questions to authors
- I would suggest citing https://arxiv.org/abs/2005.12649 and https://arxiv.org/abs/2006.09065 as they provide additional motivation for the necessity to consider more structured classes of nonconvex games.
- A popular extension beyond convexity is pseudo-monotonicity (see for instance https://arxiv.org/abs/1807.02629). How does strict hidden convexity relate to strict pseudo-monotonicity? Does the former imply the latter? Are there counterexamples?
- Do the authors expect that methods that converge in the bilinear case such as extragradient, symplectic gradient adjustment, or competitive gradient descent can be guaranteed to converge even for weak hidden convexity?

minor comments:
- Statement of Lemma 1: I think the statement would be more clear if \Sigma_1 and \Sigma_2 were defined before the conclusion.
- Theorem 2: "is a safe" -> "is safe"
- "Hidden convex-concave games & Regularizaiton" -> "...Regularization"

=============================================================================================================
After author discussion: After discussion with the authors, I am now convinced that any applicability of the theory proposed in this work to GANs is fundamentally tied to univariate latent space or generator output since the "hidden convexity assumption" does not allow for a multivariate set of latent variables to be combined to a multivariate set of outputs.
I still find the theoretical findings and method interesting, but I think that the work requires substantial refocusing and the identification of more examples of "hidden strong convexity" before being published at a top-tier conference. I therefore change my rating from 7 to 5 and recommend rejection, for now.

---

> ### Author Response · Authors · 2020-11-16
> **Response to AnonReviewer3 (1/1)**
>
> We thank the reviewer for pointing us to the interesting prior work of (Letcher 2020) and (Hsieh et al. 2020). We commit to adding a discussion in our literature review and the proposed citations.
>
> $\textit{AnonReviewer3}$ also wanted to know the connection between the pseudo-monotonicity framework of Mertiikopoulos et al. based on a variational inequality (VI) encoding of solutions to saddle point problems and the hidden convexity structural condition in our paper. The VI approach does not suffice to encode hidden convex-concave games.
>
> E.g. let’s consider a simple HCC game $\displaystyle\min_{x_1, x_2} \max_y $ $x_1x_2 y$ .
>
> In this case, it is easy to see that its saddle points are exactly the points such that at least two of $x_1,x_2, y$ are equal to zero. However, it is easy to see that e.g. the all-zero solution does not satisfy the corresponding VI which would be $x_1 x_2 y \ge 0$ for all $x_1, x_2, y$. Moreover, that framework and the analysis in that paper do not allow for equilibrium selection arguments, which is a key contribution of our work showing that a specific subset of all saddle points, i.e., the von Neumann solutions have desirable stability properties.
>
> The main concern of review was the disjointness of the parameter vectors in functions $f_i,g_j$, and its connection with the Generative Adversarial Networks. As our well-tailored and detailed work shows, even this simple case includes multiple new challenges that should be tackled in order to aim at further generalizations. We believe that our theoretical techniques may extend to more general settings, and although these questions are beyond our current scope, we hope that our work will enable this kind of interesting follow-ups.
>
> Finally, $\textit{AnonReviewer3}$ asked us how other optimistic methods in min-max optimization perform in the case of the structured hidden convex-concave games. We would like to refer the reviewer to $\textbf{Section 8.2.2}$ where we actually show how our proof technique can prove that a variation of Hamiltonian Gradient Descent achieves convergence to the von-Neumann solution problem. This already shows that the fundamental tools presented in this work are potentially applicable to other optimization dynamics as well. We believe that this is an interesting direction for future work.

---

> > ### Comment · AnonReviewer3 · 2020-11-19
> > **Does it cover GANs, or not?**
> >
> > *The main concern of review was the disjointness of the parameter vectors in functions
> > , and its connection with the Generative Adversarial Networks. As our well-tailored and detailed work shows, even this simple case includes multiple new challenges that should be tackled in order to aim at further generalizations. We believe that our theoretical techniques may extend to more general settings, and although these questions are beyond our current scope, we hope that our work will enable this kind of interesting follow-ups.*
> >
> > I reiterate my question from above: In what sense is the theory applicable to at least an idealized version of GAN with latent dimension larger than 1? Section 4 creates the impression that GANs fall into the hidden convex-concave setting but I have my doubts about that.
> > Are there other interesting classes of games that have a hidden convex-concave structure?

---

> > > ### Author Response · Authors · 2020-11-21
> > > **Comments on GAN application**
> > >
> > > Firstly, we would like to thank the reviewer for his valuable feedback.
> > >
> > > When we view GAN applications as a generator outputting a distribution and a discriminator outputting a classifier, then we agree that our results directly capture the case of “one latent dimension” for both the generator and discriminator in its full generality.
> > >
> > > In our work, we also highlight an application of WGANs that goes beyond the scope outlined above, namely parameter matching over transformed distributions. In this setting the generator outputs the parameters of a distribution instead of the distribution itself as a pdf. For the case of linear discriminators our results fully capture this case regardless of the number of parameters that need to be learned. More concretely, the min-max objective in this case is:
> > >
> > > $\min_{\theta}\max_{u} (E_{X\sim\Phi(\mathcal{D}(\theta))}[X]-E_{X\sim\Phi(\mathcal{D}(\theta^\star))}[X])u -u^2/2$
> > >
> > > Here $\Phi$ is the nonlinear smooth transformation. $\mathcal{D}(\theta)$ is the distribution of the output of the generator and can have any number of parameters that in turn can depend on any combinations of variables in $\theta$. $\mathcal{D}(\theta^*)$ is the target distribution which is realized for some setting of parameters $\theta^*$.
> > >
> > > This clearly goes beyond the “latent dimension one” case in the reviewer’s terminology. Notice that this is still a HCC with the disjoint parameters property since the generator only needs to output the difference of expectations directly and not the individual parameters. In our opinion, we find highly encouraging the fact that the subset of HCC games studied in this work can accurately predict the behaviour of WGANs proposed in prior, independent work. We hope that this addresses your question satisfactorily.
> > >
> > > Finally, we do want to point out that our goal is not to capture all aspects of GANs but to create a theoretical model that significantly pushes the boundary of CC games in the direction of GANs that still allows for theoretical analysis. As you point out HCC will make for a nice test bed for algorithms such as extra-gradient whereas previous works only explored their performance in (pseudo)monotone cases. Maybe and unlike CC they might allow for a more fine grained comparison between techniques that are convergent in all CC games but eg may require distinct types of safety conditions in HCC.

---

> > > > ### Comment · AnonReviewer3 · 2020-11-21
> > > > **Comments on GAN application**
> > > >
> > > > It is not obvious to me what the $L, F, G, f_i, g_i$ are in the example you just gave, could you please define them explicitly?

---

> > > > > ### Author Response · Authors · 2020-11-21
> > > > > **Clarification #1**
> > > > >
> > > > > The hidden strictly convex-concave game is $ L(x,y)=xy -y^2/2 $.
> > > > >
> > > > > The non-convex operators $F,G$ are:
> > > > >
> > > > >  $F(\theta)=(E_{X\sim\Phi(\mathcal{D}(\theta))}[X]-E_{X\sim\Phi(\mathcal{D}(\theta^\star))}[X])$
> > > > > and $G(\phi)=\phi$
> > > > >
> > > > > Notice that the distribution $G(\theta)$ can have an arbitrary number of parameters
> > > > >
> > > > > (e.g moments, mean,variance,scale,etc) that are controlled by an arbitrary combination of parameters in $\theta=(\theta_1,\cdots,\theta_K)$.
> > > > >
> > > > > Thus semantically, this is not a latent-dimension one example.

---

> > > > > > ### Comment · AnonReviewer3 · 2020-11-21
> > > > > > **Thanks for the clarification, one follow-up question**
> > > > > >
> > > > > > Thanks for the clarification, but it is still not clear what the $f_i$ are in this case?

---

> > > > > > > ### Author Response · Authors · 2020-11-22
> > > > > > > **Clarification #2**
> > > > > > >
> > > > > > > First, we would like to note that we changed the notation in our previous responses in order to avoid the overloading of the letter $G$ referring to both the distribution family of the generator and the operator of the discriminator. We are sorry if this caused any confusion.
> > > > > > >
> > > > > > > In this new notation, we have:
> > > > > > >
> > > > > > > $\min_{\theta}\max_{u} (E_{X\sim\Phi(\mathcal{D}(\theta))}[X]-E_{X\sim\Phi(\mathcal{D}(\theta^\star))}[X])u -u^2/2$
> > > > > > >
> > > > > > > $\bullet$ The hidden strictly convex-concave game is $ L(x,y)=xy -y^2/2 $.
> > > > > > >
> > > > > > > $\bullet$ The non-convex operators $F,G$ are:
> > > > > > >  $F(\theta)=(E_{X\sim\Phi(\mathcal{D}(\theta))}[X]-E_{X\sim\Phi(\mathcal{D}(\theta^\star))}[X])$
> > > > > > > and $G(\phi)=\phi$
> > > > > > >
> > > > > > > Notice that the distribution $\mathcal{D}(\theta)$ can have an arbitrary number of parameters
> > > > > > > (e.g moments, mean,variance,scale,etc) that are controlled by an arbitrary combination of parameters in $\theta=(\theta_1,\cdots,\theta_K)$.
> > > > > > >
> > > > > > > For instance, $\mathcal{D}(\theta)$ could be the Gaussian family of $\mathcal{N}(\mu,\sigma^2)$.
> > > > > > > We can parametrize the mean and the variance of the variables of the vector $\theta$, i.e
> > > > > > > there exists two functions $\mu=\mu(\theta),\sigma^2=\sigma^2(\theta)$. Notice that we don't impose any restriction on these functions to depend on separate variables.
> > > > > > >
> > > > > > > Indeed, in this example the operator $F$ is one dimensional so $F(\theta)=f_1(\theta)$. The aim of this WGAN example
> > > > > > > even with one dimensional $F$ we can formulate games with practical applications that involve continuous distributions. If we consider
> > > > > > > as ''latent-dimension'' the dimension of $F$ then the above HCC indeed corresponds to a latent-dimension one.
> > > > > > >
> > > > > > > On the other hand as in many machine learning applications,
> > > > > > > if we consider the parameters of the distribution $\mathcal{D}$ as the latent variables that need to be determined through
> > > > > > > moment matching, then the latent-dimension can be arbitrarily large.
> > > > > > >
> > > > > > > We would like to add that in the example above, we chose $G(\phi)=\phi$ for simplicity.
> > > > > > > Our results cover the cases of the discriminator using, in general, any smooth $G$ operator.
> > > > > > >
> > > > > > > We hope that we resolved any ambiguity.

---

> > > > > > > > ### Comment · AnonReviewer3 · 2020-11-23
> > > > > > > > **But in this example the loss depends only on expectation. GAN examples seem to be tied to one-dimensionality**
> > > > > > > >
> > > > > > > > You are right that in this example the distribution can have multiple parameters, but this is only true because the loss function only depends on a single parameter, as you said the operator $F$ has one-dimensional output.
> > > > > > > >
> > > > > > > > Based on our discussion, I am not convinced that the method has applications to GANs beyond settings that have been restricted to be essentially one-dimensional.
> > > > > > > >
> > > > > > > > As I said in my review I think that the theoretical results and techniques are interesting, but I believe that the paper should be rewritten to provide more convincing examples/motivation for the setting considered by the theory. This revision is too substantial to be done in a single round of reviews, which is why I lowered my score to 5 and recommend rejection at this time

---

> > > > > > > > > ### Author Response · Authors · 2020-11-25
> > > > > > > > > **An additional application of our framework**
> > > > > > > > >
> > > > > > > > > Given the restrictions of maximum 8 pages, in the main paper we focused our discussion on simple GAN applications that our framework can capture. However, our paper is not written with a sole focus on GANs (no explicit mention of GANs on the title and abstract) but on this new class non-convex non-concave zero sum games and our non-local convergence results, that based on our discussion the reviewer clearly appreciates. In our Appendix 9.3, we already cover a different non-GAN application that provably resolves the cycling issues in the case of the model of Vlatakis et. al (Neurips 2019). We show this problem can be effectively resolved by using regularization. We believe that it is straightforward to add a paragraph showcasing this non-GAN application of our framework since the technical analysis is already included in the appendix. We believe that an application that resolves a key problem in a Spotlight Neurips 2019 paper suffices as a contribution. We hope that in light of this, the reviewer would revert back to their original evaluation. We look forward to hearing back from the reviewer and we thank them for the engaging discussion.

---

### Author Response · Authors · 2020-11-16
**Thank you for your reviews**

We wholeheartedly thank the reviewers for the positive and encouraging feedback, as well as for the insightful comments to improve our submitted work. We continue by addressing the comments of each reviewer separately.

---

### Decision · Program_Chairs · 2021-01-07
**Final Decision**

**Decision:**

Reject

**Comment:**

This paper studies the convergence of gradient descent ascent (GDA) dynamics in a specific class of non-convex non-concave zero-sum games that the authors call "hidden zero-sum games". Unlike general min-max games, these games have a well-defined notion of a "von Neumann solution". The authors show that if the hidden game is strictly convex-concave then vanilla GDA converges not merely to local Nash, but typically to this von Neumann solution.

The paper received four high quality reviews and was discussed extensively during the author rebuttal phase. From an application angle, the authors' replies did not convince the reviewers on the relevance of this paper to GANs, and one of the original "accept" recommendations was downgraded to a "reject" because of this. On the theory side, the novelty over Vlatakis-Gkaragkounis et al. (2019) is not clear and the reviewers found the writing often confusing or hard to connect with practice. The reviewer with the most positive recommendation did not champion the paper post-rebuttal. In the end, the consensus was that the work shows significant promise, but it requires refocusing before appearing at a top-tier conference.